# The Role of Silver Nanoparticles in Electrochemical Sensors for Aquatic Environmental Analysis

**DOI:** 10.3390/s23073692

**Published:** 2023-04-02

**Authors:** Irena Ivanišević

**Affiliations:** Department of General and Inorganic Chemistry, Faculty of Chemical Engineering and Technology, University of Zagreb, Marulićev trg 19, 10000 Zagreb, Croatia; ivanisevic@fkit.hr

**Keywords:** silver nanoparticles, chemical reduction, voltammetric sensors, amperometric sensors, environmental analysis

## Abstract

With rapidly increasing environmental pollution, there is an urgent need for the development of fast, low-cost, and effective sensing devices for the detection of various organic and inorganic substances. Silver nanoparticles (AgNPs) are well known for their superior optoelectronic and physicochemical properties, and have, therefore, attracted a great deal of interest in the sensor arena. The introduction of AgNPs onto the surface of two-dimensional (2D) structures, incorporation into conductive polymers, or within three-dimensional (3D) nanohybrid architectures is a common strategy to fabricate novel platforms with improved chemical and physical properties for analyte sensing. In the first section of this review, the main wet chemical reduction approaches for the successful synthesis of functional AgNPs for electrochemical sensing applications are discussed. Then, a brief section on the sensing principles of voltammetric and amperometric sensors is given. The current utilization of silver nanoparticles and silver-based composite nanomaterials for the fabrication of voltammetric and amperometric sensors as novel platforms for the detection of environmental pollutants in water matrices is summarized. Finally, the current challenges and future directions for the nanosilver-based electrochemical sensing of environmental pollutants are outlined.

## 1. Introduction

Today, nanotechnology represents a growing multidisciplinary sector which has brought a revolution in modern science, enabling materials of specific size, structure, and composition to be formed [1,2]. Nanodimensional structures (size domain less than 100 nm) represent a bridge between atomic and bulk matter, and possess novel physicochemical properties, high surface-to-volume ratio, high adsorption and catalytic capacity, advantageous properties not present in their macro-sized counterparts [3]. The remarkable advances and tunable attributes of nanomaterials have led to increasing attention to their applications in fundamental biological research, clinical diagnostics, food safety, and environmental monitoring.

Nowadays, environmental pollution is considered a major threat worldwide, especially to the aquatic ecosystem [4,5]. With the accelerated industrialization, increasing agricultural activities, and rapid urbanization, various toxic and poisonous substances can enter natural waterways, contaminate drinking water, and subsequently cause severe health problems. Among them, heavy metal ions, nitrogen-containing inorganic species, pharmaceuticals, hormones, and nitroaromatic compounds are particularly dangerous, because of their high toxicity, carcinogenicity, and low biodegradability [6,7,8]. To date, experts have developed distinctive analytical methods for the detection of different water pollutants, including gas chromatography/mass spectrometry (GC/MS), high-performance liquid chromatography (HPLC), atomic absorption spectroscopy (AAS), and inductively coupled plasma-optical emission spectroscopy (ICP-OES) analysis, which are used to detect water pollutants [9]. While the above methods have the advantage of being very sensitive, they are also expensive and require trained personnel, making them unsuitable for on-site analysis. To solve the above problems, nanostructured materials have been recognized as powerful analytical tools for the development of facile analytical strategies for assaying trace-level aquatic environmental pollutants, with the possibility of their quantification even at femtomolar levels [10].

Over the recent decades, a variety of nanomaterials have been employed in fabrication of electrochemical sensors aiming to improve their analytical features [11,12]. The selection of functional electrode-modifying material may overcome the common analytical problems of bare electrodes, such as electrode fouling and overlapping of the redox-potential in the simultaneous determination of multiple analytes. In this context, silver nanoparticles (AgNPs) are one of the most relevant and frequently used materials for the development of versatile electrochemical sensing platforms due to their unique physicochemical properties [13]. AgNPs can be produced in an easy and cost-effective way with various morphologies, have an adjustable architecture, and are stable at room temperature. The versatility of nanosilver enables its application in a wide range of fields, from medicine (drug delivery, imaging, chemotherapeutics, gene therapy) [14], the textile industry (odour control) [15,16,17], and the food industry (safety and quality) [18], to supercapacitors and energy storage devices [19]. AgNPs also have unique catalytic properties (degradation of small organic molecules) [20,21] and superior electrical conductivity (conductive inks for printed electronics) [22,23,24]. Due to the effect of surface plasmon resonance (SPR), which is very sensitive to the particle morphology and the relative permittivity of the surrounding medium, AgNPs possess unique optical properties and can, therefore, be used as outstanding colorimetric, fluorimetric, and surface-enhanced Raman spectroscopy (SERS) nanoprobes [25,26,27,28,29]. The electrocatalytic activity in conjunction with excellent conductivity makes AgNPs a superior material for electrode surface decoration, providing a multiform base for the development of high-performance electrochemical sensors [30].

Recently, a large number of review articles have been reported regarding nanomaterial-enhanced electrochemical devices for monitoring various pollutants in the aquatic environments—heavy metals [31], chemicals [32], organic molecules [13], nutrients [33], nitrites and nitrates [8], and pharmaceuticals [4]—providing a broad survey through the field of nanosilver-based sensing platforms. This can be evidenced by the exponential growth in the number of scientific papers implementing nanosilver particles for the development of electrochemical sensors in the period of the last twenty years (determined by searching the WoS platform using the keywords electrochemical* (title) AND *silver OR Ag AND *nano AND environmental*; Figure 1). Recently, Abbas and Amin [30] thoroughly summarized scientific articles published during 2010–2020, highlighting wide spectrum of techniques available for modification of electrodes with AgNPs and their application in electroanalysis, while Zahran et al. [13] provided a comprehensive overview on AgNPs-based electrochemical sensors for the detection of aquatic organic pollutants. However, a systematic study focusing exclusively on AgNPs-based devices for screening both inorganic and organic aquatic pollutants has not yet been presented.

This review presents and discusses the current knowledge on electrochemical sensors based on silver nanomaterials for the detection of emerging pollutants in aquatic environments, summarizing the scientific papers published in the last past 5 years (since 2017). The relationship between the different morphologies of AgNPs obtained by the most commonly used reduction method (i.e., chemical, biological, and electrochemical conversion of silver(I) into nanometallic species) [34,35,36,37] and the analytical performance of voltammetric and amperometric sensors is highlighted. A review of recent advances in electrochemical sensing based on pristine AgNPs and hybrid composite architectures in conjunction with support materials, i.e., carbon-based materials (single- and multiwalled carbon nanotubes, graphene and its derivatives, carbon nitrides, carbon black, carbon cloth, graphite), metal–organic frameworks, conductive polymers, and molecularly imprinted polymers, is structured according to classes of emerging inorganic and organic aquatic pollutants. Importantly, this review provides a comprehensive evaluation from material selection and synthesis to sensor design and application of voltammetric and amperometric devices based on AgNPs for various analytes. Finally, particular challenges and research opportunities for the future development of AgNPs-based voltametric and amperometric platforms, the advantages and disadvantages of various sensors, and future development trends are given.

## 2. General Methods for the Synthesis of Silver Nanoparticles

The general mechanisms for the synthesis of silver nanoparticles can be roughly divided into two main approaches: *bottom-up* and *top-down*. The first synthetic route relies on building nanoparticles from molecules or atoms, and the second one depicts a process in which the bulk material is being mechanically broken down to the nanoscale. Both approaches play a very important role in nanoscience and nanotechnology, and can be subdivided into diverse branches, including physical, chemical, photochemical, biological, and microwave-based techniques.

### 2.1. Bottom-Up Synthesis

Atoms and molecules represent elementary building blocks for controlled *bottom-up* assembly into a nanostructured material. Therefore, this synthetic route is also referred to as the *building-up* approach [38]. Several *bottom-up* methods in solutions have the same denominator—a controlled reduction of Ag^+^ ions from a silver precursor species to elemental silver. This process can be carried out using appropriate chemical compounds or biomolecules as reducing agents, or with the help of an external source such as electrical current. Thus, these are the basic principles of chemical, biological, and electrochemical preparation methods.

The formation of functional NPs in solution is often explained by a two-stage procedure: (i) the formation of the nuclei and (ii) the growth of the nanograins [39]. The metal atoms, products of reduction reactions, when generated are insoluble in the liquid. In local regions of high supersaturation aggregation of atoms occurs, forming clusters called embryos. As a system with large free energy, these primary particles are thermodynamically unstable and subjected to coalesce into larger cluster formations—nuclei. The growth process involves the deposition of elemental units on the surface of the nuclei either by diffusion of the atoms from the solution or aggregation of nuclei clusters into bigger ones, until the final product is reached. Smaller particles are generally the result of chemical conversion with strong reducing agents; larger particles are often achieved by a slow reaction. Thus, the symbiotic role of thermodynamic and kinetic aspects plays a crucial role in determining size and shape of the prepared nanoparticles. To produce a stable dispersion of metallic nanoparticles, the use of stabilizers is essential. These agents not only protect the nanoparticles from sedimentation, but also play an important role in controlling size and shape. The *bottom-up* approach is more suitable than the *top-down* approach for fabricating uniform particles of distinct size, shape, and structure. In most cases, this technique renders spherical nanoparticles, but with altering the thermodynamic and kinetic control during the synthetic procedure it is possible to fabricate tubes, needles, rods, wires, capsules, hollow, pyramidal, and flower-shape nanostructures [40]. In the scientific articles presented in this review, the synthesis of AgNPs was carried out using the above reduction methods in solutions. Accordingly, the chemical, biological, and electrochemical preparation of AgNPs will be described in more detail.

#### 2.1.1. Chemical Reduction

Wet chemical synthesis is the predominant method used for the preparation of silver nanoparticles [41]. Pioneering work was done in 1951., when J. Turhevich reported the synthesis of gold nanoparticles by reduction of chloroauric acid with trisodium citrate [42]. *Turhevich’s method* has later been adopted for the preparation of silver nanoparticles [43]. 

For successful synthesis of silver nanoparticles in a batch, a metal salt as nanoparticle precursor, a reducing agent, and an encapsulating agent must be present. Since silver has a relatively large electropositive reduction potential (*E*° = +0.799 V), a wide spectrum of different chemicals can serve as successful reductants. Inorganic compounds such as hydrazine [44,45], sodium borohydride [46,47], and hydroxylamine hydrochloride [48], as well as organic citrate species [49,50], can be used to prepare stable colloidal suspensions. Regarding the synthesis medium, reduction processes have been studied both in aqueous solutions [51] and in organic media [52]. Polyol synthesis represents a particular route of preparation in which a polyol molecule serves as both a reducing agent and a supporting medium [53]. Acetate [54] or perchlorate [55] salts of silver may be employed as nanoparticle precursors, but silver(I) nitrate [51,56] is by far the most commonly used due to its low cost and chemical stability. Polymeric compounds [57], simple ions [58], and surfactants [59] have been reported in the literature as common nanosilver protectants. They adsorb on the surface of nanograins via covalent (chemisorption) or noncovalent (physisorption) interactions and provide separation of AgNPs by electrostatic repulsion, steric hindrance, or combined electrosteric mechanisms. Batch synthesis is an easy-to-use and cost-effective process that does not require state-of-the-art equipment (Figure 2). In general, a slight change in the synthesis route, starting from the choice of silver precursor, reducing species, and stabilizing agent, to the variation of their molar ratios, temperature or pH of the reaction medium, and duration of the chemical reduction, can cause large variations in the morphology, size, and size distribution of the nanosilver particles [60].

#### 2.1.2. Green Methods

The vast majority of methods used to prepare AgNPs cannot be considered as eco-friendly processes. The use of hazardous chemicals, as well as large amounts of waste organic solvents, pose a potential threat to the ecosystem. To solve this problem, scientists have directed nanosilver production toward the development of green chemistry synthetic approaches. Three requirements must be provided for this type of synthesis: usage of aqueous solutions, implementation of environmentally benign reducing agents, and selection of biogenic nanograin stabilizers (Figure 2). Most biological processes occur under ambient temperature and pressure, which marks green synthesis as an energy-saving approach [61].

To prepare AgNPs from silver salts, the utilization of bacterial [62], fungal [63], and algal [64] organisms as reducing and/or stabilizing compounds have been explored. A major drawback of synthesis supported by living organisms is the complex procedure and equipment required to maintain bacterial or fungal cell cultures, which also makes this method impractical for large-scale production. On the other hand, plants are widely distributed, easily available, and have a broad spectrum of metabolites that act as reducing and stabilizing agents. Thus, various plant parts in the form of seeds, leaves, roots, or flower extracts have been successfully applied in the fabrication of different size and shape nanosilver particles [65,66,67,68,69]. Despite the difficulty in obtaining a monodisperse colloid solution with pure natural sources, the biogenic synthetic route using plant extracts is the most commonly used biological fabrication method [70].

#### 2.1.3. Electrochemical Synthesis

Electrochemical synthesis represents a logical method for the production of AgNPs, since the chemical conversion of ionic silver species to zero-valent metal is a one-electron involving reaction. In electrochemical synthetic pathways electric current acts as a reducing agent. This means that the size of the nanoparticles can be precisely controlled by adjusting the current density. In addition, compared to the *bottom-up* strategies mentioned above, there is no need to introduce additional toxic chemicals into the reaction mixture. This means that the particles produced are of high purity, and the electrochemical approach is consistent with the principles of green chemistry [71]. The general electrolysis cell, consisting of reference and sacrificial electrode, serves as a reactor for the fabrication of AgNPs (Figure 3a). In brief, the electrochemical process begins with the oxidative dissolution of the silver anode, whereupon silver ions migrate through the bulk solution toward the cathodic zone. At the surface of the cathode, the ions are reduced and zero-valent silver nanoparticle seeds are formed. In the bulk solution, the stabilized nanoclusters are subjected to the Ostwald ripening process, until the final silver nanoparticle is formed [72].

In voltammetric and amperometric sensors, the functional nanomaterial is usually structured on the bare or chemically modified electrode. Thus, AgNPs can be synthesized in situ, by applying a fixed potential, current, or even cyclic voltammetry to the working electrode in a three-electrode electrochemical station (Figure 3b) [73,74]. During this approach, the applied potential (constant, ramped, or pulsed), current density, precursor concentration, and deposition time are key factors in controlling particle size, shape, and homogeneity over the active sensing surface. Electrodeposition can be performed by immersing the electrode substrate (disc or 3D electrodes) into the silver salt solution [75]. In addition, electrodeposition can be carried out from various background electrolytes such as acetic acid(AcOH)/NH_3_ buffer [76], H_2_SO_4_ [77], HNO_3_ [78], KNO_3_ [79] and phosphate buffer solution (PBS) [80]. For two-dimensional sensors (screen-printed carbon electrodes, inkjet-printed electrodes, doped rigid glass substrates), electrodeposition can also be done by drop casting the Ag(I) solution on the surface of the working electrode and applying a CV in a given timeframe [81]. This is the most suitable method for modifying planar working surfaces. 

## 3. Application of Silver Nanoparticles in Voltammetric and Amperometric Sensors

An electrochemical sensor is a device that consists of two basic parts: a receptor that specifically recognizes a substance of interest, and a transducer that converts the chemical response into a signal detectable by modern electrical instruments. Regarding the way in which the transduction process occurs, electrochemical sensors can be divided into amperometric, voltammetric, potentiometric, and conductometric [82].

The performance of all electrochemical sensors is strongly influenced by the working electrodes. In general, the redox reaction of the electroactive species at the bare electrodes (metal electrodes, glassy carbon electrodes, screen-printed or paste (carbon) electrodes, inkjet-printed electrodes) is often difficult due to the slower kinetics of the electrochemical reactions for a number of compounds and overpotential requirements. To improve the sensing mechanism, bare electrodes are usually modified with nanoengineered materials that exhibit excellent electrical conductivity and high catalytic activity such as silver [6]. Silver nanoparticles have been widely demonstrated as ideal candidates for sensing applications. Ideally, particles with a smaller size provide a larger fraction of silver atoms accessible to reactant molecules, making them a promising material for electrochemical sensors. However, AgNPs with a zero-net charge tend to coalesce into larger clusters with a lower surface-to-volume ratio. Therefore, to preserve the unique thermal and electrical properties of AgNPs, significant progress has been made towards the synthesis of AgNPs with controllable morphology, dimensions, surface charge, and physicochemical properties. To improve their sensing properties, AgNPs are often integrated into nanocomposite materials, e.g., by alloying with some another metal [83], embedded into single- [84] or multiwalled [74] carbon nanotubes, anchored on functionalized graphene (oxide) platforms [85,86], modified to form multifunctional ternary systems [87], or deposited in the form of a thin film on the electrode surface [88].

A current trend in the sensor field is directed toward solving analytical problems by developing low-cost, miniaturized, and portable devices that could be operated in the field. Electrochemical techniques, placed in order to achieve simple and sensitive analysis, are of particular interest because of their fast response and low detection limit (LOD). This chapter states the most prominent publications regarding silver nanoparticle-based sensors for monitoring of water pollutants, highlighting the relationship between the AgNP synthesis method and the voltammperometric sensing mechanism.

### 3.1. Working Principles of Voltammetric and Amperometric Sensing Techniques

In general, three-electrode systems are employed for selective voltammetric and amperometric detection of environmental pollutants, which includes a working electrode (WE) where oxidation/reduction (OR) of the electroactive species (analyte) takes place, a reference electrode (RE) that provides a constant potential, and a counter electrode (CE) that is important for a complete circuit for the charge-transfer process (Figure 4). The counter electrode is usually a platinum wire, and the reference electrode is either a standard calomel electrode (SCE) or saturated (sat.) Ag/AgCl electrode [89]. The electrodes are immersed in aqueous primary electrolyte solution to improve ionic conductivity and reduce migration during mass transfer of one or more targeted species. Voltammetric and amperometric sensors are two commonly used electrochemical devices and are, therefore, described in more detail below.

The working principle of voltammetry is based on the variance of the time-dependent excitation potential (linear, in pulses, or in squares), and the current output is correlated with the concentration of the analyte (Figure 4) [5]. In voltammetry, the measured current is composed of the Faradic current and the charging current (also known as non-faradic component). The Faradaic current results from electron transfer between the analyte and the electrode surface, and is directly proportional to the concentration of the target species. The charging current is not an analytical signal; it is the result of the perturbations in the electrode potential that lead to the charging/decharging of the electrical double layer. Cyclic voltammetry (CV) is one of the most important and widely used voltammetric techniques, where the current output is the result of a linear potential change (in the form of a triangular wave) within a given potential window. The peak current for a reversible system with diffusion control is described by the Randles–Ševčik equation (Equation (1)) [90]:(1)$$I_{p}=\left(2.69\times 10^{5}\right)n^{3/2}AD^{1/2}C\nu^{1/2}$$
where *I*_p_ stands for peak current (A), *n* is the number of electrons exchanged in the reaction, *A* is electroactive area of the electrode (cm^2^), *D* is the diffusion coefficient (cm^2^ s^−1^), *C* is the concentration of the electroactive molecule in the electrolyte (mol dm^−3^), and *ν* is the scan rate (V s^−1^). In this Equation, the constant of 2.69 × 10^5^ has units of C mol^−1^ V^−1/2^. Another technique in which the current is determined by a linearly varying electrode potential, this time in one direction, i.e., it is ideal for studying irreversible redox reactions, is linear sweep voltammetry (LSV). The other way to apply the potential is to use a pulse method—a series of pulses with a linear baseline (differential pulse voltammetry, DPV) or a square pulse amplitude applied to a stair–step waveform (square wave voltammetry, SWV). The common denominator of the stripping techniques is the preconcentration/removal of the analyte on and off the WE probe by controlling the applied voltage stress for highly sensitive quantification of the target molecule. Due to the analyte preconcentration, a much lower LOD is achieved compared to other electroanalytical methods. The deposition step can be done in three different ways: (i) by applying a negative voltage in anodic stripping voltammetry (ASV), (ii) by applying a positive voltage in cathodic stripping voltammetry (CSV), and (iii) by adsorption of the analyte in adsorptive stripping voltammetry (AdSV).

In amperometry, a constant potential is applied to the WE, and the current is measured as a function of time (Figure 4) [91]. Operating in this mode, the recorded currents can be averaged over a longer period of time, which allows more accurate quantification, i.e., amperometric sensors display high sensitivity and selectivity as well as a wide detection range. In chronoamperometry (voltage applied in steps to WE), the current output obtained is related to the bulk analyte concentration via Cottrel Equation (Equation (2)) [92]:(2)$$I=\frac{nFAD^{1/2}C}{\pi t}$$
where *I* corresponds to the diffusion current (A), *n* is the number of the electrons involved in the redox reaction, *F* is the Faraday constant, *D* represents the diffusion coefficient (cm^2^ s^−1^), *C* is the concentration of the electroactive species (mol dm^−3^), and *t* is the total electrolysis time (s). Amperometry is ideal for detection of analyte(s) in flow systems (complex matrices), as well as a highly sensitive platform for rapid analysis of targeted molecule in a simple solution. The portability of amperometric sensors also makes them ideal for in-field studies. 

Voltammetric and amperometric measurements provide a comprehensive understanding of the electrochemical behavior, i.e., the reversibility of the redox reaction, the stability of the analyte, and the kinetics of electron transfer at the electrode/electrolyte interface. An overview of the silver nanoparticle-based voltammetric/amperometric sensors for the detection of inorganic and organic water pollutants is presented for each class of compounds.

### 3.2. Electrochemical Sensors for Detection of Heavy Metal Ions

The heavy metals (HMs) are a group of naturally occurring metals and metalloids whose density is five times greater than that of water and which are toxic even in trace amounts. Among them, mercury (Hg), lead (Pb), cadmium (Cd), chromium (Cr), and arsenic (As) are biologically nonessential elements and are classified as priority metal pollutants. On the other hand, copper (Cu) is an important component of various enzymes in humans and aquatic organisms, but has toxic effects at high concentrations. Extensive anthropogenic activities (industrial and domestic use of raw materials and metal-containing products) have resulted in widespread environmental pollution. The major problem with HMs in the environment is that they are not subject to biodegradation [93], but rather bioaccumulate in different tissues by complexing with various protein functional groups. Therefore, the release of hazardous metal species into the environmental waters from industrial or municipal waste has become a global concern.

With the implementation of the Water Framework Directive (WFD) for water quality standards, the EU made the first move to reduce the harmful effects of HMs on the contamination of natural water resources [94]. New thresholds for HMs contamination in tap water have been legislated for 2020, while the Drinking Water Directive (DWD) was established in 1998 to regulate the quality of water for human consumption [95]. Therefore, the development of novel and highly sensitive sensing platforms for on-site quantification of hazardous metal contaminants in drinking and environmental water matrices is of extreme importance.

Among the electroanalytical detection methods for HMs, anodic stripping voltammetry is the most commonly used. In brief, the stripping technique involves two basic processes: (i) a deposition step, i.e., the accumulation of the analyte at the electrode surface (Equation (3)), and (ii) a stripping step, i.e., the reoxidation of the concentrated analyte on the electrode vicinity (Equation (4)). When the measurement conditions are constant, the obtained values of the oxidation peak currents are proportional to the concentrations of the analyte.
(3)$$M_{\left(\mathrm{aq}\right)}^{n+}+ne^{-}⇄M_{\left(s\right)}^{0}$$
(4)$$M_{\left(s\right)}^{0}⇄M_{\left(\mathrm{aq}\right)}^{n+}+ne^{-}$$

#### 3.2.1. Electrochemical Sensors for Divalent HMs

*Mercury (Hg)*. Mercury is a pervasive, persistent, and extremely toxic pollutant that enters the environment through gold mining and coal power plants. Long-term exposure to mercury in humans causes irreversible damage to the kidneys, central nervous system, endocrine system, and cardiovascular respiratory system. Mercury is equally toxic in all three possible oxidation states (0, +1, and +2); however, because Hg(II) has the highest water solubility, the development of sensitive and selective methods to monitor the presence of this particular species is of great public health and environmental importance [96]. Therefore, this chapter summarizes voltametric and amperometric probes based on nanosilver material for the detection of divalent mercury.

A simple, environmentally friendly and disposable voltammetric platform for the detection of Hg(II) using folic acid (FA)-stabilized AgNPs was proposed by Eksin and coauthors [97]. FA (vitamin B9) can effectively bind to the AgNPs surface via amine groups and generate a negative particle charge via the carboxylic moiety, which provides electrostatic particle stabilization. Electrochemical characterization of the pencil graphite electrode (PGE) improved with FA-AgNPs was performed by CV and electrochemical impedance spectroscopy (EIS) techniques, while DPV was used for quantitative identification of the analyte (detection limit of 8.43 μM under the optimized conditions) (Figure 5a). The main advantage of the FA-AgNPs-PGE sensing platform is the time savings; only 2 h are required to perform the entire procedure—from electrode preparation to analysis. A dual optical and electrochemical sensor for rapid sensing of toxic mercury based on *Agaricus bisporus* (AB) synthesized AgNPs was proposed by Sebastian et al. [98]. In optical detection, a visible change in the color of the suspension was followed by a decrease in the absorption maxima, indicating particle aggregation by Hg(II)-AgNP-AB complexation as a highly selective mercury sensing mechanism. The same approach was used in the electrochemical sensing, where a reversible redox couple was formed in the presence of mercury in acetate buffer solution (ABS, pH = 6.0). The enhancement of the current signal is favored by the presence of small AgNP-AB (mean diameter of 14.13 nm), which significantly increases the effective surface area of the platinum electrode (PE) substrate. The analytical performance of the AgNPs-AB/PE sensor manifested in well-defined DPV plots (scan rate of 100 mV s^−1^), with peak current linear to mercury at micromolar concentration. Another dual sensor for the detection of Hg(II) in water specimens based on biosynthesized AgNPs was fabricated by Punnoose et al. [99]. The AgNP-MD (*Mimosa diplotricha*)-modified Pt electrode exhibited a well-defined redox pair at the potentials of +336 mV and +80 mV, which was attributed to the Ag^+^/Ag conversion. In the presence of the analyte, the appearance of two prominent current peaks at lower potentials, i.e., a high anodic through (293 mV) and an apparent reduction peak (70 mV), indicated excellent nanosilver sensing activity towards mercury. At an optimal state (0.1 M acetate buffer, pH = 6.0, pulse amplitude 25 mV, and pulse width 200 ms), the DPV responses to various Hg(II) concentrations were linear within the micromolar concentration range, with a regression coefficient of 0.9990 and a detection limit of 1.46 μM. The designed green sensor successfully detected Hg(II) in tap water and river water specimens with RSD of 0.6–1.8%. In addition to voltammetric sensing, the biosynthesized nanosilver material also showed catalytic activity towards degradation of the dyes methyl orange (MO) and methylene blue (MB). Another strategy to alter the hydrophylicity-lipophilicity of AgNPs and, thus, improve their interactions with hazardous mercury is to encapsulate them with cyclophane moieties, especially calixarene and their derivatives [100]. A colorimetric and amperometric sensor for the detection of mercury in aqueous solution was prepared by modifying a Pt wire with thiophene-functionalized calix[4]-arene-capped AgNPs (ThC-AgNPs) [101]. The electrochemical behavior of the prepared sensing surface towards Hg(II) was investigated by CV measurements in PBS and revealed a cathodic peak at +0.92 V, ascribed to the reduction of Hg(II) to its monovalent state. The linear dependance of the peak current on the square root of scan rate confirmed the Faradic nature of the reaction, with very low nonfaradic current present in the sensing system. The amperometric *I*–*t* curve, recorded by successive addition of the analyte in 0.1 M PBS (pH = 5.8) at an applied potential of −0.035 V, revealed that the steady-state current is reached within a few seconds, promoting its applicability for real-time performance (Figure 5b). The decrease in peak current was shown to be proportional to analyte concentration in the range of 0.9–9 μM, with a detection limit of 10 nM. George and Matthew [102] developed a Hg(II) selective colorimetric, fluorescent, and electrochemical sensor based on AgNPs, produced and stabilized with *Curcuma longa* rhizome extract. The electrochemical response of the CLR-AgNPs/AuE (gold electrode) voltammetric sensor was studied by CV and DPV analysis. In the absence of the analyte, well-defined CVs were observed at +0.12 V and −0.07 V associated with Ag^+^/Ag^0^ conversion, while the new peaks occurred at +0.16 V and −0. 31 V due to the complex of the curcumin stabilizer with Hg(II) ions. The Hg(II) was selectively detected in the presence of Cd(II), Co(II), Cr(III), Cu(II), Mn(II), Ni(II), Pb(II), and Zn(II); no noticeable impact of additional HMs on the detection of mercury was observed. Overall, the study confirmed that the green AgNPs are a very sensitive multisensing probe for the detection of Hg(II) with LODs of 8.20 nM (colorimetry), 0.50 nM (fluorescence), and 6.50 nM (voltammetry), respectively. AgNPs stabilized with two biocompatible molecules—carboxymethylcellulose (CMC) and polyvinylpyrrolidone (PVP)—were deposited on the glassy carbon electrode (GCE), and the effect of particle morphology on the analytical features of the voltammetric sensor for trace-level detection of Hg(II) was investigated [103]. The usage of two capping agents affected: (i) the crystallite morphology—octahedral CMC@AgNPs and spherical core-shell PVP@AgNPs structures were fabricated; (ii) different values of interfacial resistance—0.7 kΩ for PVP@AgNPs/GCE and 14 kΩ for CMC@AgNPs/GCE, and consequently (iii) lower LOD for CMC-encapsulated AgNPs (0.19 nM) compared to PVP-capped particles (73 nM), determined by differential pulse anodic stripping voltammetry (DPASV). The accuracy of the proposed sensor was also tested using the AAS method, with lower values obtained using the electrochemical platform. The Compton group used a simple and reproducible method to detect aqueous Hg(II) ions by galvanic displacement of silver by mercury, resulting in the loss of the stripping signal associated with AgNPs [104]. The large surface area of citrate-capped AgNPs drop casted onto the GCE allows highly sensitive analytical performance, as only small amounts of hazardous pollutants are required to displace measurable amounts of silver. Employing LSV (sweeping the potential from −0.01 V to +1.0 V against a SCE; scan rate 0.1 V s^−1^), current maxima were linear within a wide concentration range (100.0 pM–10.0 nM) and a detection limit of 28 pM was achieved.

*Cadmium* (*Cd*). Cadmium is a heavy metal often referred to as the metal of the 20th century, because of its widespread use in industry—galvanizing and electroplating, rechargeable Ni-Cd batteries, electrical conductors, and in the manufacture of alloys, pigments, and plastics. Cd is released into the environment through anthropogenic sources such as smelting and mining, and is considered as a prevalent environmental contaminant [105]. High exposure to cadmium causes severe damage to kidneys, lungs, and liver, and increases the risk of cancer. Therefore, many researchers are focusing of the development of selective and sensitive methods to quantify cadmium in the environmental and drinking water. An extract of *Allium sativum* (garlic) has been shown to be an excellent green agent for the synthesis of AgNPs and for the removal of hazardous cadmium [106]. Divalent cadmium induces aggregation of the AgNP-AS; a mechanism successfully used in optical sensing (color change), fluorescence response (enhancement of fluorescence intensity with increasing cadmium concentration), and electrochemical sensing (prominent CVs in the presence of the analyte). DPV current outputs (recorded in 0.1 M ABS) were linear with analyte concentration in the range of 10–90 μM, corresponding to a detection limit of 0.277 μM cadmium.

*Lead (Pb)*. Lead is a highly poisonous metal found in the Earth’s crust [107]. The widespread use of lead in households and industry has resulted in extensive contamination of water sources caused by the perennial deterioration of paints, corrosion of metal water pipes, and leaching of ceramic containers used in the manufacture of lead-containing batteries. Thus, the significant environmental damage, human exposure, and serious public health issues require the development of rapid and efficient methods for Pb detection. Ganash and Alghamdi [108] proposed a sensitive and selective strategy for the electrochemical detection of lead ions based on a nanocomposite modified carbon paste electrode (CPE). The WE was prepared by mixing polyaniline (PANI), graphite, and paraffin oil, and the green-synthesized AgNPs were drop casted onto the electrode surface. Compared with bare CPE, PANI/CPE, and AgNPs/CPE, the prepared AgNP/PANI/CPE exhibited the lowest charge transfer resistance (*R*_ct_) and the highest conductivity. The electrochemical detection was performed in three steps: (i) accumulation at open-circuit potential (8 min); (ii) preconcentration step at a potential of −0.3 V (100 s) vs. saturated Ag/AgCl; and (iii) application of SWV technique (ABS, pH = 5.5, in the potential range of −0.7 V to −0.2 V) to oxidize zero-valent lead. This simple sensor exhibited good recoveries (104%, 110%, and 116% for wastewater, biologically treated wastewater, and tap water samples, respectively, assessed by spiking with known Pb concentration) and excellent reproducibility (RSD values of less than 10% obtained for other HMs). Highly crystalline Ag@Pt core-shell nanoparticles for selective detection of lead in environmental water samples were synthesized by a bioinspired method, using extract of *Psidium guajava* leaves, ascorbic acid, and microwave irradiation [69]. The prepared nanocatalyst (14.5 nm core diameter and 4.55 nm shell thickness) was successfully decorated on a plain graphite paste electrode (GPE). Compared to the bare electrode, the implementation of the nanohybrid material enlarged the effective surface area (from *A* = 0.066 cm^2^ to *A* = 0.145 cm^2^) and resulted in faster electron transfer (58% reduction in the electron transfer resistance). To achieve ultratrace detection, the analyte was first electrodeposited at a potential of −1.2 V vs. sat. Ag/AgCl for 300 s, after which SWASV was performed (0.1 M acetate buffer, pH = 5; stripping potential of −0.43 V). The fabricated sensor successfully detected lead in the sewage water sample (0.48 μM), and the result was further verified using AAS with a deviation of only 3.0%. Amino acid synthesized NiO-Ag particles with ball-flower (BF) morphology were found to be highly selective for lead ions due to their large active surface area and high conductivity [109]. The usage of L-glutamine was pivotal for the synthesis and anisotropic growth of the nanocomposite. Namely, the formation of functional flower structured BF-NiO-Ag particles was initiated by the complexation of metal ions with carboxyl groups, and the generation of steric particle stabilization by amino acid side chains. Detection of lead ions in an electrolyte buffer solution (0.1 M ABS, pH = 6.0) was performed by DPV; the dependence of the reduction peak current was found to be linear with Pb(II) concentrations ranging from 1.5 nM to 10 nM (Figure 5c). The analytical application of the BF-NiO-Ag/GCE sensor was investigated in pipe water and groundwater samples with satisfactory average recoveries and relative standard deviations, while selectivity was confirmed in the presence of 100-fold transition metal ions.

*Copper* (*Cu*). Copper is one of the main essential trace metals in the human body and a key component of several enzyme systems. At higher concentrations, copper can be toxic and cause liver damage, gastrointestinal disorders, and neurodegenerative diseases [110]. The bark extract of *Moringa oleifera* was utilized as an efficient reducing and stabilizing agent to produce AgNPs that serve as multifunctional sensing probes (optical, fluorescent, and electrochemical) for hazardous Cu(II) in environmental waters [111]. A simple coating technique was used to modify WE, and the efficiency of the AgNP-MO/PE sensing platform towards divalent copper was validated with CV (sharp current output only in the presence of Cu(II); a flat current line was obtained in aqueous solutions of Hg^2+^, Zn^2+^, Cr^3+^, and Ni^2+^, respectively), and DPV (linear increase in redox current with increase in Cu(II) concentration) techniques. Besides sensory application, the biosynthesized AgNPs also showed antibacterial activity towards *Escherichia coli* (gram-negative) and *Staphyloccocus aureus* (gram-positive) extracted from real water samples. Recently, Zhou and coworkers developed a selective Cu(II) sensor based on 4-mercaptobenzoic acid (MBA)-capped AgNPs and fluorine-doped tin oxide (FTO) decorated with gold nanoislands and MBA [47]. The peculiarity of this sensor lies in the indirect mechanism of copper quantification through the AgNPs-MBA-Cu^2+^-MBA-AgNPs carboxylate complex architecture, where the role of MBA-AgNPs is to amplify the electrochemical oxidation signal. Under the optimized conditions (potential window from −0.15 V to +0.15 V at a scan rate of 100 mV s^−1^), the SWV peak currents were proportional to the logarithmic value of copper concentration in the range from 0.1 nM to 100 nM. The quantitative copper sensing assay was tested in tap water and pool water samples, and the accuracy and reliability were also quantified using ICP-MS.

**Figure 5 sensors-23-03692-f005:**
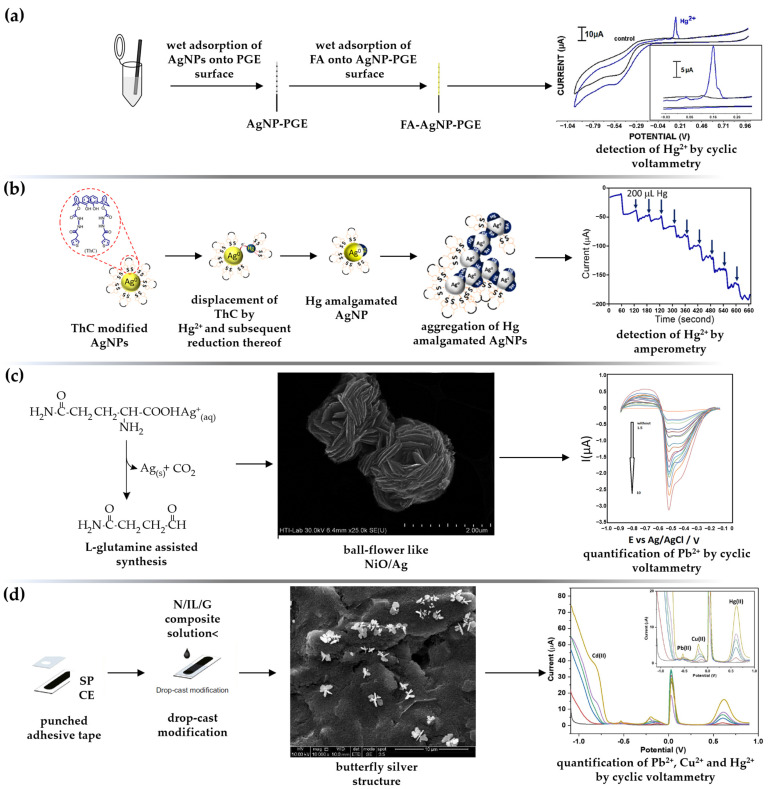
Illustration of AgNPs-based voltammetric and amperometric sensors for single and simultaneous detection of HMs in aquatic medium. (**a**) Fabrication steps of FA-AgNPs/PGE voltammetric sensor. In brief, PGE was firstly immersed in sodium citrate buffer solution of biosynthesized AgNPs and further immersed in folic acid (FA) solution for CV detection of Hg(II). Adapted from Ref. [97], copyright by John Wiley and Sons. (**b**) Amperometric sensor based on AgNPs displacement by divalent mercury. Adapted from [101] with permission. Copyright under ACS AuthorChoice License. (**c**) Pb(II)-selective DPV sensor based on ball-flower-like NiO/Ag/GCE sensing platform. Reproduced from Ref. [109], with permission from the Royal Society of Chemistry. (**d**) DPASV sensor based on SPCE decorated with butterfly-shaped silver nanostructure for the simultaneous detection of Cd(II), Pb(II), Cu(II), and Hg(II) in tap water and environmental (rain, lake, river) water, respectively. Reproduced from Naseri et al. [79] under the terms of Creative Commons CC BY license.

*Simultaneous detection of divalent heavy metal ions*. In general, simultaneous detection of multiple ionic species with the same probe is a complex process, because overlap of the current peaks due to the similar redox potential values may occur [112]. Therefore, to obtain optimal voltammetric responses to HMs, it is of utmost importance to choose the optimal technique, as well as to optimize various experimental parameters. In contrast to the detection of single HM species, almost all nanosilver-based sensors presented in this review employ anodic stripping techniques for the simultaneous HMs detection.

Perez-Rafols and coworkers [113] synthesized two types of AgNPs with different size and morphology—Ag nanoseeds (*d* = 9.1 ± 0.4 nm) and Ag nanoprisms (*d* = 14.0 ± 0.9 nm)—via borohydride and citrate reduction, paired them with three different carbon substrates (graphite, graphene, and carbon nanofibers), layered over a screen-printed carbon electrode (SPCE), and used them to simultaneously detect copper and lead in groundwater matrices with certified metal value. The most favorable analytical performance (highest selectivity) was achieved with the nanoseed crystallites/carbon nanofiber adjusted SPCE. Under the optimal stripping parameters (deposition potential *E*_d_ = −1.1 V, deposition time *t*_d_ = 120 s, acetate buffer pH = 4.5), the as-prepared voltammetric Ag-nanoseed-SPCNFE platform exhibited satisfactory analytical features—a low simultaneously detected LOD value (1.98 μg/L for lead and 2.99 μg/L for copper) and good reproducibility (RSD value of 2.94% and 1.2% for lead and copper, respectively). To increase the sensor surface area and electrocatalytic activity towards simultaneous detection of Cu and Pb, metal–organic frameworks (MOFs) have been incorporated into the voltammetric sensor design [114]. In general, MOFs—a crystalline network of metal ions and organic ligands—are highly porous, thermally stable, and have various functional groups that can uniformly hold functional metal nanoparticles at high density, resulting in an increase in active sensor surface area [115]. The three-dimensionally ordered crystalline architecture was spread on a 1 × 1 cm^2^ indium tin oxide (ITO) glass substrate, and the prepared Ag@MOF/ITO sensor exhibited remarkable sensitivity in voltammetric detection of HMs, with LODs of 0.68 μM (copper) and 0.64 μM (lead), respectively [114]. An electrochemical deposition method was used to fabricate polyrutin/AgNPs/GCE sensor for voltammetric determination of Pb(II) and Cd(II) in water and soil samples [77]. To obtain a highly selective sensor, the WE was first immersed in 0.1 M ABS to preconcentrate the target HMs via the complex alloy formation on the electrode surface. Preconcentration potential of −1.2 V was applied for 240 s for the deposition step. In DPASV, zero-valent metals were oxidized to divalent cations by scanning the potential from −1.3 V to +0.3 V. Increment in HMs concentration resulted in an increase of the corresponding anodic stripping peak current in the nanomolar linear concentration range and a detection limit of 10 nM (*S*/*N* = 3) was achieved for both analytes. At the same time, the detection process was not interfered by other ions, and the target ions can be analyzed even in the presence of high concentrations (100-fold excess) of F^−^, SO_4_^2−^, NO_3_^−^, SO_4_^2−^ anions, and Na^+^, Ca^2+^, Mg^2+^, K^+^, Al^3+^, Cr^2+^, Fe^2+^, and Ba^2+^ cations.

Hassan et al. [73] presented a voltammetric sensor for the simultaneous determination of three divalent HMs—cadmium, lead, and copper—using GCE coated with poly(1,8-diaminonaphthalene) (p-1,8-DAN) conductive polymer, and with electrodeposited AgNPs. The simultaneous determination of HMs by ASV was investigated in the voltage range of −0.20 V to 0.0 V (vs. Ag/AgCl) with an amplitude of 0.025 V, an accumulation time of 120 s, and a scan rate of 0.005 V s^−1^, in different concentration ranges of the analyte. The remarkable differences in the stripping peak potentials between cadmium (−1.02 V), lead (−0.78 V), and copper (−0.32 V) obtained with the AgNPs@p-1,8-DAN/GCE sensor, compared to the GCE coated only with a polymer layer, demonstrated the role of AgNPs in the enhancement of current responses. The nanoprobe was tested in tap water samples spiked with a certain amount of HMs, and recoveries between 103.0% and 108.0% were achieved. The combination of high electrical conductivity and high area-to-volume ratio of graphene nanoplates (GrNPs) and excellent catalytic activity of AgNPs was used to fabricate an electrochemical platform for simultaneous voltammetric detection of divalent Cd, Cu, and Pb in tap water [78]. AgNPs/GrNPs were deposited on the graphite electrode (GE), and the electrochemical behavior of the AgNPs/GrNPs/GE sensor was preliminarily characterized by CV technique using the 5.0 mM [Fe(CN)_6_]^3–/4–^ redox probe (1:1) in a 0.1 M KCl primary electrolyte solution. A pair of prominent OR peaks were detected (high current maxima and low peak-to-peak separation), indicating the advantages of both the nanosilver and the graphene material. Using the SWASV technique (*E*_d_ = −1.2 V, *t*_d_ = 200 s, voltage range from −1.0 V to 0.0 V vs. sat. Ag/AgCl; in 0.1 M ABS, pH = 5.0), the stripping peaks of the targeted HM ions appeared at different potentials (−0.78 V for Cd, −0.57 V for Pb and −0.21 V for Cu, respectively) with significant peak separations. The prepared voltammetric sensor displayed great stability and the possibility of long-term use—94%, 93%, and 91.5% of the initial response was retained after 7 weeks of storage at room temperature (RT). A working electrode made of triethylenetetramine (TETA) functionalized rGO nanosheets decorated with small-sized AgNPs (10–20 nm) was used as a simple but extremely sensitive voltammetric platform for the simultaneous detection of hazardous Cu, Cd, and Hg [85]. The introduction of TETA was beneficial in several ways: (i) it promoted the reduction of GO sheets forming the amide bond with the oxygen-rich functionalities; (ii) lowering the oxygen content improved the graphene conductivity; (iii) it prevented rGO stacking by maintaining the sheet spacing; and (iv) it enabled the incorporation of AgNPs. The distinctive feature of this hybrid sensor architecture lies in its analytical properties—detection limits below femtomolar concentrations were achieved by combining the usage of a simple instrumentation (two-electrode electrochemical workstation) and a simple quantification technique (CV).

Different voltammetric platforms have been developed for the simultaneous detection of four different HMs, i.e., lead, cadmium, copper, and mercury, one based on AgNPs/rGO nanocomposite as interface material layered over a magnetic GCE [116], and the other based on a butterfly-shaped silver nanomaterial modified SPCE [79]. In the first report [116], the SWASVs exhibits four distinct and well-defined peaks (−0.74 V for Cd, −0.56 V for Pb, −0.05 V for Cu, and +0.32 V for Hg, respectively), and a wide linear range from 0.05 μM to 3.5 μM for all target ions with LODs of 0.254 μM (Cd), 0.287 μM (Pb); 0.171 μM (Cu) and 0.180 μM (Hg) were achieved in the simultaneous analysis. In the second report [79], DPASV was employed for simultaneous detection of four target HMs in tap water, rainwater, lake water, and river water. Under the optimal conditions (0.1 M acetate buffer, pH = 4.4; *E*_d_ = −0.7 V; *t*_d_ = 30 s), individual peaks occurred at −0.788 V, −0.536 V, −0.209 V, and +0.627 V, corresponding to the oxidation of cadmium, lead, copper, and mercury, respectively (Figure 5d). This simultaneous platform was successfully tested in real water samples and quantified with ICP-OES analysis, confirming the accuracy and reliability of the proposed sensor.

#### 3.2.2. Electrochemical Sensors for Trivalent and Hexavalent HMs

*Arsenic (As)*. Arsenic is a metalloid that occurs naturally in minerals, usually in association with sulfur and metals (arsenopyrite, cobaltite, enargite, etc.). Due to the complex combination of natural processes (soil erosion, volcanic eruptions, bacterial decomposition) and anthropological activities (mining activity, usage of arsenic-containing pesticides), various forms of arsenic circulate in the environment through geological cycle (air, soil, and water). In its two inorganic forms, i.e., trivalent arsenites and pentavalent arsenates, arsenic is highly toxic; however, As(III) has been shown to be more harmful and more difficult to remove from water sources [117]. Long-term exposure to arsenic-containing water can lead to adverse health problems, from skin lesions to many types of cancer. Considering its toxicity, detection of arsenic is extremely important. Recently, electrochemical methods, especially stripping voltammetry, have made great progress in detection trace amounts of As(III) in water samples [118].

The citrate-stabilised AgNPs dispersed on the surface of the Au electrode were proposed as an effective tool for the selective electrochemical determination of arsenic in river water [50]. Compared to the bare Au electrode, the voltammogram recorded on the modified electrode generated a well-defined peak at a potential of −0.28 V, which was attributed to the three electron reduction of As(III) to a zero-valent form. The DPASV currents (recorded in 0.1 M HNO_3_; *E*_d_ = −0.6 V; *t*_d_ = 5 min) were linear with increasing As(III) concentrations in the micromolar range (0.05–0.2 μM), with a detection limit of 0.0138 μM. To improve analytical performance in arsenic detection, polypyrrole nanowire (PpyNW) was coated with AgNPs, sandwiched between a pair of gold electrodes, and used as a selective sensing platform [119]. As(III) was detected by the two-step stripping voltammetry technique: (i) reduction of the As(III) at the electrode surface by enrichment of As(0) at a potential of −0.3 V for 10 s, respectively, and (ii) reoxidation of As(0) from the electrode surface (PBS, pH = 4.0), resulting in a current response directly proportional to the analyte (0.01–0.10 μM concentration range). Compared to the previously reported AgNPs/Au sensor [50], the detection limit of the PpyNW/AgNPs electrode was lowered to the ppb region [119]. The excellent plasmonic properties and unique electrochemical activity of AgNPs were used as a sensing strategy to fabricate a multimodal arsenite sensing assay [120]. For this purpose, AgNPs were first coated with asparagine (Asn), and then modified with reduced glutathione (GsH) and dithiotreitol (DTT). The effect of the capping agent on the binding ability of As(III) was investigated; it was found that the AgNPs encapsulated with Asn were insensitive to the analyte, whereas GSH-AgNPs showed a weak signal. Due to the abundant oxygen and sulphur-containing groups, the GSH/DDT/Asn-AgNPs probe interacts with As(III), resulting in aggregation of the probe in solution or on the surface of the gold electrode, boosting both the optical and electrochemical signals. From the DPV results, the linear working range for the concentration of As(III) was between 0.01 ppb and 40 ppb, with a LOD of 5.2 ppt (*S*/*N* = 3). In addition, the concentration of As(III) measured with the dual-mode sensor agreed well with the results of the standard ICP-MS method.

*Chromium (Cr)*. This transition metal can occur in several oxidation states between 0 and VI, with the trivalent and hexavalent states being the most abundant and, thus, the forms most commonly encountered in the environment and in industrial settings [121]. Cr(III) is an essential trace element, but it can be harmful to humans in high concentrations. On the other hand, the carcinogenic Cr(VI) is one of the most concerned toxic heavy metals due to its extensive use in industries (leather tanning, electroplating). The hazards of Cr(VI) are reflected not only in cell damage (it possess strong ability to penetrate biofilms, disrupt the DNA transcription process, and cause chromosomal aberrations and cell apoptosis), but also in bioaccumulation through the food chain. Hence, it is a pertinent issue to effectively monitor hexavalent chromium concentrations to ensure the safety of the environmental and drinking water sources.

AgNPs synthesized from *Lycopersicon esculentum* have been used as an effective material for selective optical, fluorescent, and electrochemical sensing of Cr(III) in lake water and water from metal plating industry [122]. This wild tomato species was selected as an optimal green synthesizing agent because it is rich in proteins and organic acids (silver reducing and stabilizing biomolecules), as well as citrate anions (chromium chelating agents). Optimal CV performance of the AgNP-LE/Pt platform for analyte measurement was achieved in 0.15 M ABS (pH = 12.5); current outputs were linear with scan rate, indicating a surface-confined electron transfer reaction. Therefore, DPV was chosen for the quantification of Cr(III), and broad linear performance vs. chromium concentration (*R*^2^ = 0.9963) was achieved with a detection limit of 0.804 μM. For the simultaneous determination of Cr(III) and Cr(VI) species (detected by different analytical techniques), bimetallic gold-silver (oxide) nanoparticles were fixed on the surface of SPCE [81]. Detailed X-ray photoelectron spectroscopy (XPS) analysis showed that the freshly prepared sensor contained only elemental silver and gold, however, spontaneous oxidation of both metals occurred after one day. Negative voltage (*E* = −0.2 V) is applied for the Cr(III) pre-concentration step, after which DPV analysis (pulse potential of 0.05 V, pulse time of 0.08 s, potential window of −0.2 V to +0.3 V at a step potential of 0.008 V) was performed. Under optimum conditions (0.1 M ABS, pH = 4.5; *E*_d_ = 0.6 V, *t*_d_ = 120 s), the SWV responses to hexavalent chromium concentration were linear within the ppb concentration range. This dual-channel electrochemical device was successfully used to detect both chromium oxidation states in tap water, artificial saliva, and artificial sweat samples. To fabricate an electrochemical Cr(VI) sensing platform, electrodeposition was performed from a 0.20 M AcOH/NH_3_ (pH = 5.9) supporting electrolyte solution containing 0.05 mM AgNO_3_ to obtain an Ag-plated GCE [76]. Since the voltammetric signal of Cr(VI) is notably affected by the properties of the silver sensing film, the effects of deposition potential, concentration of silver precursor, and deposition time were studied in detail. DPASV was proposed as an electrochemical method for the quantification hexavalent chromium, with current peaks being linearly proportional in the range of 0.35–40 μM. The detection process of the in situ prepared Ag-plated GCE sensor was not interfered by Na^+^, K^+^, SO_4_^2−^, CO_3_^2−^, HCO_3_^−^, and H_2_PO_4_^−^; however, the presence of Fe^2+^, Fe^3+^, and chloride (at concentrations greater than 0.12 mM) significantly decreased the current output. A novel and facile one-pot approach, combining silver nitrate precursor, 4,4′-dihydroxy biphenyl (BP) as a hydrogen bond donor, and planar 4,4′-biphenoquinone (BPQ) as a hydrogen bond acceptor, was introduced for the first time to for the synthesis of AgNPs anchored on BP-BPQ nanoribbons (NRs) [123]. The hybrid material prepared was then mixed with graphite and parafine oil to produce GPE as a sensing platform for indirect determination of hexavalent chromium. The modification resulted in higher conductivity and larger surface area of the sensor. In addition, the AgNPs serve as the active sites for the selective accumulation of Cr(VI) compounds, resulting in higher sensitivity. Under the optimal conditions (0.1 M PBS, pH = 2), HCrO_4_^–^ is the dominant Cr(VI) form that triggers the decrease in DPV current of the BP/BPQ redox system. The fabricated AgNPs-2D-biphenol-biphenoquinone sensing platform exhibited a low detection limit (2.0 × 10^−12^ M) and good recoveries for spiked Cr(VI) concentrations in tap water, river water, and electroplating waste water matrices. Table 1 summarizes scientific articles on various electrodes decorated with nanosilver material as sensor assays for voltammetric and amperometric detection of HMs in aquatic environment.

**Table 1 sensors-23-03692-t001:** Survey of the reviewed silver nanomaterial-based electrochemical sensors for detection of heavy metal ions in water matrices.

Analyte	Sample	Sensor Design/ Detection Method	Linear Range	AgNP Synthetic Approach	LOD	Refs.
Hg(II)	Tap water	AgNP-FA-PGE/CV	10–25 μM	Green synthesis	8.43 μM	[97]
Hg(II)	Lake water	AgNP-AB-Pt/DPV	10–90 μM	Green synthesis	2.1 μM	[98]
Hg(II)	Tap water, river water	AgNP-MD-Pt/DPV	5.0–45 μM	Green synthesis	1.46 μM	[99]
Hg(II)	Tap water, drinking water	ThC-AgNPs/Pt/Amp	45–105 nM	Chemical reduction	10.0 nM	[101]
Hg(II)	River water, well water	CLR-AgNP/AuE/DPV	0.01–0.06 μM	Green synthesis	6.50 nM	[102]
Hg(II)	Tap water, river water, lake water	CMC@AgNPs/GCE/DPASV	5.0–75 μM	Chemical reduction	0.19 nM	[103]
Hg(II)	Environmental water, drinking water	AgNPs/GCE/LSV	100.0 pM–10.0 nM	Chemical reduction	28.0 pM	[104]
Cd(II)	Lake water, water from pigment, cosmetics and fertilizer industries	AgNP-AS-Pt/DPV	10–90 μM	Green synthesis	0.277 μM	[106]
Pb(II)	Tap water, wastewater	AgNPs/PANI/CPE/SWV	0.1–120 μM	Green synthesis	0.04 μM	[108]
Pb(II)	River water, tap water, sweage water	Ag@Pt/GPE/SWASV	0.25–10 μM	Green synthesis	0.8 nM	[69]
Pb(II)	Pipe water, groundwater	BF-NiO-Ag/GCE/DPV	1.5–10 nM	Green synthesis	0.06 nM	[109]
Cu(II)	Lake water, waste water from electroplating unit	AgNP-MO/PE/DPV	10–90 μM	Green synthesis	0.530 μM	[111]
Cu(II)	Tap water, pool water	AgNPs-MBA-Cu^2+^-MBA-AuNPs-FTO/SWV	0.1–100 nM	Borohydride reduction	0.08 nM	[47]
Cu(II) Pb(II)	Groundwater (certified reference material)	AgNPs-SPCNFE/ASV	7.6–130.7 μg/L 3.2–162.5 μg/L	Borohydride reduction Citrate reduction	2.29 μg/L 0.96 μg/L	[113]
Cu(II) Pb(II)	Aqueous solution	Ag@MOF/ITO/CV	1.0–50 μM 1.0–50 μM	Chemical reduction	0.68 μM 0.64 μM	[114]
Pb(II) Cd(II)	Tap water	Polyrutin/AgNPs/GCE/DPASV	29–140 nM 57–203 nM	Electrodeposition	3.0 nM 10.0 nM	[77]
Cd(II) Pb(II) Cu(II)	Tap water	AgNPs@*p*-1,8-DAN/GCE/SWASV	0.90 nM–9.0 μM 2.0 nM–24.0 μM 1.3 nM–9.0 μM	Electrodeposition	0.17 nM 0.15 nM 0.09 nM	[73]
Cd(II) Cu(II) Pb(II)	Tap water	AgNPs/GrNPs/GE/SWASV	0.5–120 μg/L	Electrodeposition	5.0 ng/L 4.1 ng/L 1.0 ng/L	[78]
Cu(II) Cd(II) Hg(II)	Aqueous solution	rGO/AgNPs/CV	1.0 nM–10 μM	Hydrothermal	10^−15^ M 10^−21^ M 10^−29^ M	[85]
Pb(II) Cd(II) Cu(II) Hg(II)	Aqueous solution	AgNPs/rGO/MGCE/SWASV	0.05–2.5 μM 0.05–3.5 μM 0.05–3.5 μM 0.05–3.0 μM	Chemical reduction	0.141 μM 0.254 μM 0.178 μM 0.285 μM	[116]
Cd(II) Pb(II) Cu(II) Hg(II)	Tap water, rainwater, lake water, river water	AgNS/SPCE/DPSV	5.0–300 ppb 5.0–300 ppb 50–500 ppb 5.0–100 ppb	Electrodeposition	0.4 ppb 2.5 ppb 7.3 ppb 0.7 ppb	[79]
As(III)	River water	AgNPs/Au/DPASV	0.05–0.2 μM	Chemical reduction	0.0138 μM	[50]
As(III)	Real water	AgNPs/PpyNW/Au/ASV	0.01–0.10 μM	Electrodeposition	1.5 ppb	[119]
As(III)	River water, lake water, well water	GSH/DTT/Asn-AgNPs/AuE/DPV	0.01–40 ppb	Green synthesis	5.2 ppt	[120]
Cr(III)	Lake water, water from metal plating industry	AgNP-LE-Pt/DPV	10–90 μM	Green synthesis	0.804 μM	[122]
Cr(III) Cr(VI)	Tap water, wastewater	Ag-Au/SPCE/DPV	0.05–1.0 ppm 0.05–5.0 ppm	Electrodeposition	0.1 ppb	[81]
Cr(VI)	Tap water	Ag-plated/GCE/DPASV	0.35–40 μM	Electrodeposition	0.10 μM	[76]
Cr(VI)	Tap water, river water, electroplating wastewater	AgNPs-BPQ-BP-NRs/GPE/SWV	1.0–100 μM 0.01–1.0 μM 0.08–10 nM	Chemical reduction	2.0 pM	[123]

AgNPs-FA—folic acid-coated silver nanoparticles; PGE—pencil graphite electrode; CV—cyclic voltammetry; AgNP-AB—*Agaricus bisporus* coated AgNPs; Pt—platinum electrode; DPV—differential pulse voltammetry; AgNP-MD—*Mimosa diplotricha* coated AgNPs; ThC—thiophene-functionalized calix[4]-arene; Amp—amperometry; CLR—*Curcuma longa* rhizome; AuE—gold electrode; CMC—carboxymethylcellulose; GCE—glassy carbon electrode; DPASV—differential pulse anodic stripping voltammetry; LSV—linear sweep voltammetry; AS—*Allium sativum*; PANI—polyaniline; CPE—carbon paste electrode; SWV—square wave voltammetry; BF—ball flower; AgNP-MO—*Moringa-oleifera-*coated AgNPs; MBA—4-mercaptobenzoic acid; FTO—fluorine-doped tin oxide; SPCNFE—carbon-nanofiber-modified screen-printed electrode; MOF—metal–organic framework, ITO—indium tin oxide; *p*-1,8-DAN—poly(1,8-diaminonaphthalene); SWASV—square wave anodic stripping voltammetry; GrNPs—graphene nanoplates; GE—graphite electrode; rGO—reduced graphene oxide; MGCE—magnetic glassy carbon electrode; SPCE—screen-printed carbon electrode; PpyNW—polypyrrole nanowire; GSH/DTT/Asn—reduced glutathione-dithiothreitol-asparagine; AgNP-LE—*Lycopersicon-esculentum-*coated AgNPs; BPQ-BP-NRs—biphenol-biphenoquinone nanoribbons.

### 3.3. Electrochemical Sensors for Nitrogen-Containing Inorganic Species

Nitrogen compounds are one of the most important nutrients in nature. They occur as part of the nitrogen cycle in soil and aquatic systems in various organic (proteins, amino acids, urea, living or dead organisms) and inorganic (gaseous nitrogen, ammonia, ammonium ions, nitrite, and nitrate) forms [91]. Due to extensive anthropogenic activities, the concentration of nitrogen-containing substances in the natural environment has increased dramatically. Regulations for the maximum allowance concentrations of nitrogen-based inorganic species are related to the specific water source. Since groundwater and surface water are very important sources of drinking water, the development of an accurate, sensitive, and low-cost platform for in situ monitoring of various nitrogen-containing species has been and continues to be of great importance. Table 2 provides an overview of the voltammperometric sensors based on silver nanomaterial for the detection of nitrogen-containing inorganic water pollutants.

#### 3.3.1. Electrochemical Sensors for Nitrite (NO_2_^−^) and Ammonium (NH_4_^+^) Detection

Nitrite anions (NO_2_^−^) are common inorganic pollutants found in soil, drinking water, and biological systems. Control of nitrite concentration is of great importance because its presence in the human body affects the rate of oxygen transport in the blood by irreversibly converting hemoglobin to methemoglobin. In addition, nitrite anions can also react with amines to produce carcinogenic nitrosamine species [124,125]. Therefore, the World Health Organization (WHO) has limited the nitrite concentration in drinking water to 65.2 μM [126]. Nowadays, there is considerable interest in the monitoring of nitrites as well as the development of a simple and reliable low-cost nitrite sensors [127]. The irreversible mechanism of electrocatalytic nitrite oxidation at the electrode modified with nanosilver material in neutral and weakly acidic solutions can be simply described by Equation (5) [128]:(5)$$NO_{2}^{-}+H_{2}O\to NO_{3}^{-}+2H^{+}+2e^{-}$$

For strongly acidic solutions, where the nitrite species is unstable, a different oxidation mechanism is proposed. However, since the environmental and tap water samples have a nearly neutral pH, reduction mechanism in acidic media is beyond the scope of this review and has, therefore, been omitted.

Aqueous extracts from leaves of *Salvia leriifolia* were used to prepare AgNPs with spherical morphology and an average diameter of 27 nm [65]. Besides the exceptional antibacterial activity, the biosynthesized AgNPs were used to fabricate a simple nitrite sensing platform by anchoring them on the surface of a GCE. The electrochemical behavior was investigated applying CV technique (0.1 M PBS; pH = 7.0), and the two prominent anodic peaks at +0.31 V (formation of Ag_2_O monolayer) and +0.78 V (oxidation of Ag_2_O to AgO), and one cathodic peak at +0.11 V (reduction of AgO to Ag_2_O) confirmed the electrocatalytic nanosilver activity. Under optimized conditions, the *S.l*-AgNPs/GCE sensor showed a linear dependence of the peak electroreduction current vs. nitrite concentration in a narrow micromolar concentration range. Since the main focus of this work was oriented towards a simple, environmentally friendly approach to obtain particles with enhanced antibacterial activity, no detailed insights into the nitrite sensing mechanism nor LOD value were provided. Hajisafari and Nasirizadeh [75] proposed a voltammetric sensor for simultaneous quantification of hydroxylamine and nitrite fabricated by decorating the GCE with electrochemically synthesized AgNPs. First, rectangular-shaped particles were prepared by electrodeposition from an acidic 1 mM AgNO_3_ solution at a continuous potential cycling from −0.7 V to 1.9 V, and a sweep rate of 80 mV s^−1^ in 7 cycles. To obtain a functional oxadiazole-modified sensor (OAgNPs/GCE), the electrode was then immersed in 0.1 mM PBS (pH = 7.0) containing 1.0 mM oxadiazole. Unlike the bare GCE, or electrodes decorated only with nanosilver material and/or oxadiazole, the fabricated sensor was able to separate the oxidation peak potentials of hydroxylamine and nitrite even in the presence of interfering species. DPV was employed as a more sensitive electroanalytical technique for quantification of targeted analytes and gave satisfactory LOD values of 57.8 μM (NH_2_OH) and 4.1 μM (NO_2_^–^), as well as excellent recoveries in drinking water and tap water samples. An amperometric nitrite sensor including GC as the core electrode material, this time decorated with green-synthesized nanosilver material, was fabricated by Shivakumar et al. [129]. Silver nanospheres (AgNS) with an average crystallite size of 30 nm were produced using waste from the paper industry as reducing liquor without any purification steps. The XRD pattern of the fabricated nanosilver material showed strong diffraction peaks at 2*θ* values corresponding to the (111), (200), and (311) planes of the face-centered cubic crystalline (FCC) metallic silver structure. The applicability of the prepared nanospheres in nitrite sensing was investigated by an amperometric experiment. The amperometric current response, obtained at an applied potential of +0.86 V in 0.1 M PBS (pH = 7.0), showed linearity in the nitrite concentration range from 0.1 μM to 8.0 μM, with a sensitivity of 580 μA/mM/cm^2^.

Graphene and graphene-derivative-based composites have also attracted considerable attention in the electrochemical detection of nitrites in water samples [8]. Core-shell Au@Ag nanoparticles (average diameter of 60 nm), synthesized on carboxylated graphene (CG) sheets and drop casted on GCE surface, provided a specific platform for dual detection of nitrite and iodide (Figure 6a) [86]. The electrochemical behavior of the target species was thoroughly investigated (0.1 M PBS, pH = 7.4), partly to exclude the formation of insoluble AgI on the surface of the nanohybrid material. The DPV technique was chosen for the simultaneous determination of the analytes because it yields distinct peaks in low concentration ranges. In the quantification of nitrite, the oxidation currents at a potential of +0.95 V increased linearly with the increase in analyte concentration, without any significant shift in peak position. For I^−^, two prominent anodic peaks occurred, one corresponding to the oxidation of iodide to triiodide (I_3_^–^) form (+0.55 V), and the other to the further oxidation of triiodide to elemental iodine. Although the presence of two iodide oxidation peaks indicates that the presented sensor is more effective for accurate detection of I^−^, it provides a broader linear range for the quantification of nitrite. Recently, Zhao et al. [130] presented a simple and effective preparation route for gamma-ray-assisted in situ synthesis of AgNPs on a graphene oxide platform (GO) attended for nitrite sensing. A complex of silver(I) salt with imidazole, incorporated into the graphite oxide wall, served as a precursor for the preparation of polyvinylpyrrolidone (PVP)-stabilized AgNPs. By changing the ratio of the reactants, the radiolytic reduction undergoes different mechanisms and a broad spectrum of GO-AgNP nanocomposites with gradient reduction degrees were obtained. A functionalized nanohybrid was deposited on the GCE surface to fabricate the nitrite sensing platform. For electrochemical characterization, a Pt wire and the Ag/AgCl reference electrode were immersed into a 0.1 M PBS (pH = 7.2) in addition to the WE, forming a three-electrode measuring system. CVs recorded at the modified electrode in 1 mM nitrite solution rendered anodic current response of 2.08 mA at a potential of 1.08 V, indicating oxidation of nitrite to nitrate ions due to the abundant oxygen-containing functional GO groups. Measurements performed with the addition of 100 μM Na_2_SO_4_, NaCl, NaNO_3_, and Na_2_CO_3_ solution showed no response to interferences. The detection limit of 0.24 μM indicated that the proposed sensor was stable, reproducible, sensitive, and selective for nitrite detection. Another simple nitrite sensing platform, composed of AgNPs and reduced graphene oxide (rGO) nanocomposite material, was proposed by Ahmad et al. [131]. Different volumes of nanocomposite suspension (i.e., 1 μL, 3μL, 5 μL, and 7 μL) were drop casted on the GCE surface to monitor the sensor fabrication steps and to examine the effect of the modifier layer thickness on the sensor performance. CV and EIS measurements revealed that the optimal sensor performance was achieved when 5 μL of Ag-rGO composite was added. GCE coated with a smaller amount of the nanocomposite exhibited higher charge transfer resistance (*R*_ct_) values and less prominent current maxima, and the sensor prepared with 7 μL of composite suspension was too thick to ensure a fast electron transfer rate. DPV was employed as a sensitive nitrite quantification technique, exhibiting high sensitivity (18.4 μA/μM cm^2^) and low detection limit (0.012 μM). Practical application of the sensor was demonstrated in the detection of nitrites in pond water with satisfactory recoveries in the range of 94.5–102.6%. Another approach to improve the conductivity and sensitivity is to use conductive polymers (CPs) in voltammperometric sensor design [132]. Biphasic interfacial polymerization using pyrrole (Py) as reducing agent was carried out to obtain silver nanoparticles of uniform size uniformly distributed on the GO sheets [133]. The AgNPs@PPy/rGO nanocomposite was layered on the surface of GCE and employed for electrochemical oxidation of hydrazine and nitrite inorganic pollutants. The [Fe(CN)_6_]^3−/4−^ redox probe was used for CV and EIS experiments. Both electroanalytical techniques showed an improvement in the sensor output after electrode modification with silver nanomaterial—an increase in the oxidation peak current for the CV measurements, and a decrease in electron transfer resistance (*R*_ct_) for the EIS experiments. Two distinctive and well-separated current peaks were observed vs. saturated Ag/AgCl reference electrode: one at +0.53 V corresponding to hydrazine oxidation, and a second one at +0.76 V describing nitrite oxidation. The potential window of 470 mV indicates that the current maxima do not overlap; the DPV method can be utilized for simultaneous electrochemical detection of both analytes in tap water, lake water, and drinking water. Excellent sensor stability was achieved after 250 cycles with a decrease in the initial sensor responses of 2.8% (hydrazine) and 1.7% (nitrite). A few years later, the same group of authors reported on voltammetric sensor for simultaneous detection of sulfite and nitrite in water samples using AgNP, rGO, and polyaniline (PANI) nanocomposite-modified GCE [134]. The bare GCE displayed a flat and poor CV response, while the GCE modified with AgNPs@PANI/rGO showed a symmetric and sharp anodic peak at a potential of +0.84 V vs. saturated Ag/AgCl electrode, indicating fast electron transfer kinetics at the sensor–nitrite system. Under optimal conditions (0.1 M PBS, pH = 7.0; scan rate 25 mV s^−1^; both pulse width and amplitude 25 mV), two distinct DPV responses with 550 mV peak potential differences were observed, and the oxidation current trough increased linearly with increasing nitrite concentration within broad micromolar concentration range. Compared to the previously presented sensor (nitrite LOD value of 21 nM) [133], this sensor exhibited lower performance at the nitrite detection limit (56 nM). This may be explained by slightly higher conductivity of the AgNPs@PPy composite compared to the AgNPs@PANI sensing platform [134]. GO, this time functionalized with ethanolamine and AgNPs, was also used as a suitable 2D nitrite sensing platform [135]. The distribution of AgNPs is affected by reaction temperature; the temperature effect on the morphology of silver nanostructure was investigated at different temperatures (80–140 °C; Δ*t* = 20 °C); scanning electron microscopy (SEM) images showed that the highest nanosilver yield and uniform distribution on the wrinkled graphene sheets was achieved at 100 °C. Compared with the bare GCE, GO/GCE, AgNPs/GCE, and AEfG1000/GCE, the Ag-AEfG100/GCE sensor exhibited highest current (8.5 μA) at the lowest oxidation potential (+0.83 V); the synergistic effect of the AgNP/AEfG composite improves electrocatalytic nitrite quantification. The results of CV were confirmed by EIS measurements, where the resistance value of the chemically modified electrode (70 Ω) was significantly lower than that of the bare GCE (140 Ω). Quantitative analysis was studied by amperometry (successive nitrite additions every 50 s in PBS, pH = 7.4) at a constant oxidation potential of +0.85 V, where the sensor exhibited a wide micromolar linear range. The prepared amperometric sensor did not response to various interferences (KCl, NaNO_3_, MgSO_4_, CaCl_2_, glucose, nitrobenzene, and *p*-nitrotoluene), and applicability in real matrices (tap water samples) with satisfactory recoveries (99.5%; 99.5% and 100.61%, respectively) was achieved. Multiwalled carbon nanotubes (MWCNTs) decorated with electrodeposited AgNPs expressed synergistic effect towards nitrite oxidation by expanding the working area of the GCE [74]. Compared to the GCE decorated solely with MWCNTs, the AgNPs/MWCNTs/GCE sensor exhibited a shift in operating nitrite oxidation potential from +0.790 V to +0.784 V upon the addition of 1 mM nitrate, indicating that employment of nanosilver material favours rapid electron transfer. The proposed sensor expressed rapid amperometric response, showing linearity over a wide concentration range (1.0–100 μM) with a sensitivity of 0.19 μA μM^−1^ and LOD value of 0.095 μM.

Besides GCE, the other forms of carbonaceous materials have also been validated as a promising tool for electrochemical detection of nitrites. Salagare and coworkers [136] performed a nonaqueous reduction of silver(I) salt with ethylene glycol (EG) to produce AgNPs decorated ZnO nanocomposite material for the selective electrochemical detection of nitrite. Under optimal conditions (*E* = +0.76 V in acetate buffer, pH = 6.0), the sensor exhibited a notable improvement in CV response to the oxidation of nitrite, with the current maxima being proportional to nitrite concentrations within a wide linear range. The novelty of the proposed sensor lies in the facile and inexpensive AgNP synthesis approach used to fabricate a nitrite-selective sensor with LOD of 14 μM. Chitosan, a chemically inert biopolymer with hydrophilic functional groups, is suitable for surface functionalization of carbon-based nanomaterials via noncovalent interactions that form water-stable complexes [137]. A chitosan-functionalized, AgNPs-modified MWCNT paste electrode (MWCNTPE) was used for the adsorption cyclic voltammetry determination of nitrite in Britton–Robinson (BR) buffer solution (pH = 4.0) [138]. Compared to other voltammetric nitrite sensors, the presented assay showed exceptional analytical performances, with the LOD value obtained at nanomolar concentrations. CPE, enriched with the natural clay mineral halloysite nanotubes (HNT), silver nanorods, and molybdenum disulfide (MoS_2_) as a composite material (Ag/HNT/MoS_2_), have been successfully applied for the detection of nitrite in water (Figure 6b) [139]. MoS_2_ is one of the most typical 2D-layered transition metal dichalogenides with functional application in electrochemical catalysis [140]. Nyquist plots of 1.0 mM [Fe(CN)_6_]^3−/4−^ redox couple, recorded in 0.1 M KCl solution for bare CPE, Ag/HNT-CPE, HNT/MoS_2_-CPE, MoS_2_-CPE, and Ag/HNT/MoS_2_-CPE, show a gradual decrease in *R*_ct_ value, which promoted electron transfer from the nitrite to the sensor surface [139]. The addition of Ag/HNT leads to the aggregation of MoS_2_ sheets, i.e., the formation of catalytically active edge sites, providing a favorable pathway for electron transport. The stepwise amperometric response (*E* = +0.80 V) obtained by the successive addition of 10 μM nitrite into the 0.1 M phosphate buffer solution (pH = 4.0) showed a wide linear concentration range (2.0–425 μM) and LOD (0.7 μM) below the WHO guideline for drinking water [126]. The recoveries of nitrite spiked into real water matrices were in the range of 96.5–99.6%, confirming the applicability of the proposed amperometric sensor for the quantification of nitrite [139]. Protonated carbon nitride (H-C_3_N_4_) decorated with AgNPs was layered on the surface of carbon cloth (CC) to provide a flexible nitrite-sensing platform [141]. The Ag/H-C_3_N_4_ hybrid material possessed higher conductivity and larger surface area per volume, which enabled contact with the analyte and promoted electron transfer. In addition, a flexible CC skeleton promoted exposure of the active sites and facilitated electron charge transfer. The amperometric Ag/H-C_3_N_4_/CC platform was selective toward nitrite with a sensitivity of 0.8577 μA/mg and with LOD value of 0.216 μM. The reliability of this nitrite sensor was demonstrated in tap water, with recoveries of 100.81% and 101.23% determinated by amperometric and UV-Vis spectrophotometry. In addition to carbon-based materials, glass can also serve as a substrate for the fabrication of the nitrite-sensing platform. Pang et al. [142] used silver nanoparticles functionalized with a poly(3,4-ethylenedioxytiophene):polystyrene (PEDOT:PSS) film as a glass working electrode for selective nitrite detection. The conductive polymer was spin-coated onto the glass surface pretreated with concentrated H_2_SO_4_ (CSA) and decorated with electrochemically synthesized nanosilver material (Figure 6c). The electrochemical oxidation of nitrite on the film-sensor surface occurs at +0.95 V, indicating that AgNPs play an important role as a catalytic active center. The Ag-CSA-PEDOT:PSS film electrode exhibited a broad linear range (from 0.5 μM to 3400 μM), and a detection limit of 0.34 μM was achieved in PBS (pH = 7.4). To evaluate the potential of the prepared sensing surface, amperometric experiments were performed in the presence of interferences (NaNO_3_, KCl, MgSO_4_, CaCl_2_, and glucose), and no response to the addition of inorganic and organic species was observed.

Compared to potentiometric sensors [143], voltammetric and amperometric sensors are not commonly used for ammonium (NH_4_^+^) quantification. Baciu et al. [144] reported a novel sensing platform based on AgNPs and CNTs for the simultaneous detection of nitrite and ammonium species in aqueous matrices. This is also the only example of a nanosilver-based voltammetric sensor for ammonium species detection published within the time window studied. The CNT-epoxy composite electrode was randomly decorated with larger silver aggregates by electrodeposition from a 0.1 M AgNO_3_ solution at a constant potential of −0.4 V/SCE; the resulting Ag-CNT sensing platform provided satisfactory electrical conductivity (0.713 S cm^−1^) for analytical applications. The oxidation process of ammonium and nitrite species studied by the CV technique in the primary electrolyte (0.1 M Na_2_SO_4_) revealed a striking oxidation peak for both analytes at +0.15 V and +0.7 V, respectively. SWV and DPV techniques were employed to evaluate the analytical performance of the prepared sensor. Under the optimized operating conditions (0.05 V step potential, 0.2 V modulation amplitude, and 0.05 V s^−1^ scan rate), DPV provided improved sensitivities for the detection of the target analytes (0.613 mA mM^−1^ for ammonium and 0.980 mA mM^−1^ for nitrite), and 1 μM (NH_4_^+^) and 0.7 μM (NO_2_^−^) LOD values were achieved.

**Figure 6 sensors-23-03692-f006:**
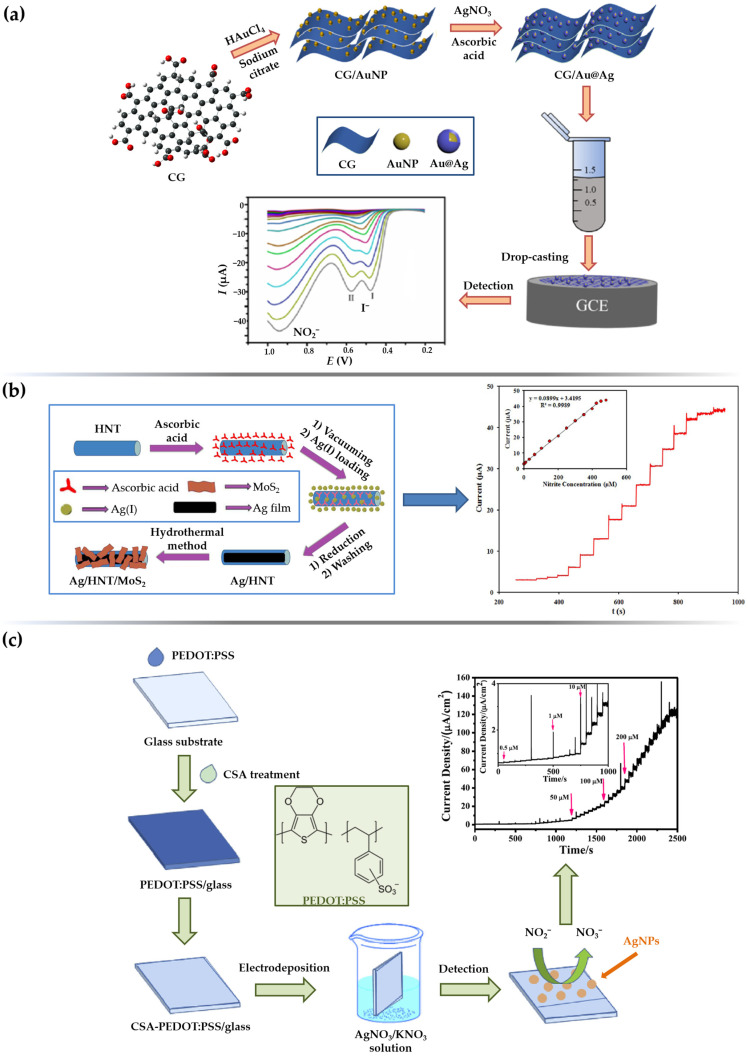
Schematic presentation of selected voltammetric and amperometric sensors for nitrite in aquatic medium. (**a**) Nitrite and iodide detection device based on bimetallic Au@Ag/carboxylated graphene heterostructure decorated over a GCE. Reprinted with permissions from [86]. Copyright 2023 Elsevier. (**b**) Fabrication steps of Ag/HNT/MoS_2_ on carbon paste electrode and corresponding *I* vs. *t* response. Reprinted from [139] with permission. Copyright 2023 Elsevier. (**c**) Fabrication of Ag-CSA-PEDOT:PSS over a glass substrate. Adapted from [142] with permission. Copyright 2023 Elsevier.

#### 3.3.2. Electrochemical Sensors for Nitrate (NO_3_^−^) and Ammonia (NH_3_) Detection

Nitrate (NO_3_^−^) is one of the most common nitrogen-containing contaminants in aquatic systems. Adverse human activities have permanently altered the global nitrogen cycle, and excess nitrate from agricultural lands leaches through the soil into groundwater, lakes, and rivers, and eventually enters into drinking water [8,145]. Excess of nitrate disrupts aquatic ecosystem dynamics and leads to eutrophication processes, water quality degradation, and biodiversity destruction. In animal species and humans, excessive nitrate intake leads to the formation of nitrosamines, and subsequently to cancer. Therefore, the development of a low-cost and portable instrument that can perform routine measurements in various water matrices is highly desirable. The most promising analytical tool that can meet these requirements are voltammetric and amperometric sensors.

The analytical performance of electrochemical nitrate sensors is highly dependent on the efficiency of the catalysts, which can be strongly influenced by the pH of the water matrices and the composition of the working electrode used [146]. In other words, the mechanism of electrocatalytic reduction of nitrates on various metallic (Pt, Pd, Cu, Au, Ag, Zn, Fe, Ru, Bi, Sn, Au(Hg), etc.) and chemically modified electrodes is a complex process that can lead to the formation of nitrites or various gaseous nitrogen species [147]. Considering that most of the environmental waters have a (near) natural pH, electrochemical reduction in neutral solution has been studied in detail, developing voltammetric and amperometric sensors using different types of nanosilver as catalysts.

A dual-modified (Pt region and Ag region) coated planar ITO substrate served as the sensing surface for the simultaneous detection of ammonia and nitrate in a simulated water sample containing analyte concentrations as in the groundwater [148]. ITO substrate was carefully divided into two closely spaced regions, and Pt particles (0.1 M HCl containing 1 mM H_2_PtCl_6_, *E* = +1.4 V for 120 s) and Ag particles (0.5 M KNO_3_ containing 5 mM AgNO_3_, *E* = −0.8 V for 20 s) were electrodeposited from aqueous solutions. A linear correlation was observed between the nitrate reduction current and the nitrate concentration in the milimolar range, with milimolar nitrate LOD. Good reproducibility of the Pt//Ag/ITO sensor versus nitrate was observed (RSD = 3.93%). Another planar substrate, i.e., a screen-printed carbon electrode modified with AgNPs and Cu(II)-terephthalate framework nanocomposite, was used as a sensing platform for amperometric nitrate detection in water and soil extracts [46]. Spherical NPs were deposited in situ over the cuboidal Cu-BDC-MOF particles, whose surfaces served as support material for the nucleation and development of AgNPs. The AgNPs/Cu-BDC-MOF/SPCE sensor performed well under optimal amperometric conditions (−0.80 V vs. saturated Ag/AgCl electrode) over a wide linear range (0.5–1000 μM), with a LOD value of 0.24 μM. The electrode was successfully applied to detect nitrate in water and soil extracts, with good recoveries in the range of 97.8–102%. A schematic presentation of the sensor design can be seen in Figure 7.

In addition to the planar ITO electrode and SPCE, the electrochemical determination of nitrate has also been successfully demonstrated using AgNPs-modified disc electrodes [146,149,150]. A stainless steel acupuncture needle was used as base for the development of electrochemical a nitrate sensing platform [150]. This microneedle electrode (MNE) was first coated with chitosan (CTS) and PVP film (electrodeposition from 0.1 mg mL^−1^ PVP and CTS solution in 0.1 M HAc, *E* = −2.0 V, *t* = 200 s), to improve the conductivity, specific surface area, and binding sites for functional nanomaterial. Then, CTS/PVP/MNE was immersed into the silver nitrate solution, and the larger AgNPs were assembled uniformly over the MNEs polymeric coating. An amperometric response of the AgNPs/CTS/PVP/MNE towards nitrite reduction was performed in 0.1 M NaCl solution (at the constant potential of −1.1 V), with current output being linear and the analyte concentration within 0.005–2 mM range (*R*^2^ = 0.996). Practical application of the proposed sensor was shown in coastal seawater samples. A mechanically (alumina) and electrochemically (polarization in 0.5 M H_2_SO_4_) cleaned gold electrode (*d* = 0.3 μm) was decorated with AgNPs (electrodeposition from 0.1 M KNO_3_ containing 0.17 mM AgNO_3_ by chronoamperomety, *E* = −0.2 V), and used for selective nitrate quantification [146]. The effective electrode surface area depends strongly on the electrical charge (*Q*) required for successful silver deposition—over increasing the charge leads to a decrease in current maxima. In other words, larger silver aggregates are formed by the Ostwald ripening process, and the catalytic efficiency of the fabricated AgNPs/Au sensor decreases significantly, resembling the behavior observed for pure Au electrodes. SWV was proposed for the quantification of nitrate in artificial sea matrices (34.5 g L^−1^ NaCl), and a linear sensor range in the micromolar region was achieved. The stability of the AgNPs/Au system showed insignificant differences in the current signal recorded over a 26-day period, indicating the applicability of the proposed voltammetric sensor for in-field nitrate monitoring in marine water. In situ functionalization of a gold substrate with AgNPs, obtained by direct decomposition of an organometallic precursor in an organic liquid phase under mild conditions, was reported for the first time as a facile method to fabricate a highly selective nitrate-sensing surface [149]. Using physical vapor deposition (PVD) technique, a 400 nm thick gold film was deposited on a boron-doped silicon wafer, resulting in a 8 mm × 8 mm WE with planar square geometry. This electrode modification provides a suitable route for electroreduction of nitrate with SWV in the potential range of −1.3 V to −0.10 V with respect to the reference SCE. Under optimized conditions (35.4 g L^−1^ NaCl artificial seawater; pH = 6.0), the maximum current of nitrate reduction at a potential of −0.8 V increases linearly with the increase of nitrate concentration. Compared to the conventional electrodeposition method, the in situ synthesis of AgNP on the Au surface by organometallic decomposition resulted in improvement in sensor performance, i.e., a three orders of magnitude lower LOD value. This is the lowest LOD value obtained not only for seawater matrices, but also for all nitrate sensors presented in this review, demonstrating that the small size, morphology, and uniform distribution of the nanosilver electrocatalyst is crucial in the development of the voltammperometric sensors.

**Table 2 sensors-23-03692-t002:** Survey of the reviewed silver nanomaterial-based electrochemical sensors for detection of nitrogen-containing inorganic water pollutants.

Analyte	Sample	Sensor Design/ Detection Method	Linear Range	AgNP Synthetic Approach	LOD	Refs.
NO_2_^−^	Water	*S.l*-AgNPs/GCE/CV	1.0–3.75 μM	Green synthesis	–	[65]
NH_2_OH NO_2_^−^	Drinking water, tap water	OAgNPs/GCE/DPV	183.4–779.2 μM 15.3–64.9 μM	Electrodeposition	57.8 μM 4.1 μM	[75]
NO_2_^−^	Tap water	AgNS/GCE/Amp	0.1–8.0 μM	Green synthesis	0.031 μM	[129]
NO_2_^−^ I^−^	Drinking water, river water, sewage water	Au@Ag/CG/GCE/DPV	2.5–1250 μM 3.5–1000 μM	Chemical reduction	0.15 μM 0.1 μM	[86]
NO_2_^−^	Water	AgNP-GO/GCE/LSV	1.0 μM–1.0 mM	^60^Co *γ*-irradiation-assisted chemical reduction	0.24 μM	[130]
NO_2_^−^	Pond water	Ag-rGO/GCE/DPV	0.1–120 μM	Microwave-assisted chemical reduction	0.012 μM	[131]
NO_2_^−^	Tap water, lake water, drinking water	AgNPs@PPy/rGO/GCE/DPV	0.6–8.6 μM	Chemical reduction	6.8 nM	[133]
SO_3_^2−^ NO_2_^−^	Drinking water, tap water, river water	AgNPs@PANI/rGO/GCE/DPV	2.7–24.4 μM 1.0–28.2 μM	Chemical reduction	77.0 nM 56.0 nM	[134]
NO_2_^−^	Tap water	Ag-AEfG100/GCE/Amp	0.05–3000 μM	Chemical reduction	0.023 μM	[135]
NO_2_^−^	Water	AgNP/MWCNTs/GCE/DPV	1.0–100 μM	Electrodeposition	0.095 μM	[74]
NO_2_^−^	Lake water, pickle water	Ag-ZnO/PGE/DPV	30–1400 μM	Chemical reduction	14.0 μM	[136]
NO_2_^−^	River water	Chit-AgNPs/ MWCNT/PE/CV	–	Borohydride reduction	30.0 nM	[138]
NO_2_^−^	Water	Ag/HNT/MoS_2_/CPE/Amp	2.0–425 μM	Green synthesis	0.7 μM	[139]
NO_2_^−^	Tap water	Ag/H-C_3_N_4_/CC/Amp	5.0–1000 μM	Green synthesis	0.216 μM	[141]
NO_2_^−^	Tap water	Ag-CSA-PEDOT:PSS/glass/Amp	0.5–3400 μM	Electrodeposition	0.34	[142]
NO_2_^−^ NH_4_^+^	Groundwater	Ag-CNT/DPV	0.2–1.0 mM	Electrodeposition	0.006 mM 0.003 mM	[144]
NO_3_^−^ NH_3_	Simulated water sample	Pt//Ag/ITO/CV	0.27–10.9 mM Nonlinear (NH_3_)	Electrodeposition	0.134 mM 3.946 μM	[148]
NO_3_^−^	Water, soil extract	AgNPs/Cu-BDC/SPCE/Amp	0.5–1000 μM	Borohydride reduction	0.24 μM	[46]
NO_3_^−^	seawater	AgNPs/CTS/PVP/MNE/Amp	5.0–2000 μM	Electrodeposition	1.2 μM	[150]
NO_3_^−^	Artificial seawater	AgNPs/Au/SWV	0.39–50 μM	Electrodeposition	–	[146]
NO_3_^−^	Artificial seawater	AgNPs/Au/SWV	0.9 nM–1000 μM	In situ deposition	0.9 nM	[149]

*S.l*-AgNPs—*Salvia leriifolia*-coated AgNPs; GCE—glassy carbon electrode; CV—cyclic voltammetry; OAgNPs—oxadiazole-coated AgNPs; DPV—dynamic pulsed voltammetry; AgNS—silver nanospheres; Amp—amperometry; CG—carboxylated graphene; AgNP-GO—silver nanoparticles-graphene oxide; LSV—linear sweep voltammetry; rGO—reduced graphene oxide; PPy—polypyrrole; PANI—polyaniline; AEfG—ethanolamine-functionalized graphene; MWCNTs—multiwalled carbon nanotubes; PGE—pencil graphite electrode; Chit-AgNPs—chitozan-coated AgNPs; PE—paste electrode; HNT—halloysite nanotube; CPE—carbon paste electrode; H-C_3_H_4_—protonated carbon nitride; CC—carbon cloth; CSA—concentrated sulfuric acid; PEDOT:PSS—poly(3,4-ethylenedioxytiophene):polystyrene sulfonate; Ag-CNT—silver–carbon nanotube; ITO—indium tin oxide; Cu-BDC—copper(II) terephthalate; SPCE—screen-printed carbon electrode; CTS—chitosan; PVP—polyvinylpyrrolidone; MNE—microneedle electrode; Au—gold electrode; SWV—square wave voltammetry.

### 3.4. Electrochemical Sensors for Phenolic Compounds

Phenolic compounds are a group of small molecules containing a hydroxyl group attached to the carbon atom of an aromatic ring. Based on their chemical structure and number of carbon atoms, phenolic compounds can be divided into several classes. Due to their toxicity and persistency, phenolic compounds are also the most polluting components in the environment, especially in water sources [6]. Therefore, numerous studies aim at simple and effective detection of phenolic compounds via designing voltammetric and amperometric sensor surfaces. Table 3 summarizes selected voltammetric and amperometric sensors for the detection of phenolic-like contaminants in aquatic environments.

Among phenol-based compounds, *phenol* is the most toxic species and is classified as a priority pollutant [151]. An eco-friendly synthetic approach using five different plant leaf extracts (*Basil*, *Geranium*, *Eucalyptus*, *Melia*, and *Ruta*) as reducing and stabilizing AgNPs agents for water quality monitoring has been reported [66]. XRD analysis revealed common Miller indices for the crystalline FCC structure, while the size of nanosilver grains, calculated by the Scherrer formula, yielded crystallites between 21 nm (AgNP-M) and 50 nm (AgNP-E). Thus, applying DPV technique, the GCE sensor modified with AgNPs-*Melia* particles was highly selective towards phenol even in the presence of bisphenol A and catechol, exhibited a LOD value of 0.42 μM and retained 93.11% of its initial response after four consecutive weeks of measurement. In another work, Zhu and Yang [80] reported an amperometric sensor based on the AgNPs/CNT for simultaneous detection of phenol and *o*-cresol in shale gas wastewater. The electrodeposition from 0.5 mM AgNO_3_ (applied potential range from −0.7 V to 1.4 V in 0.1 M PBS/1.0 M ascorbic acid) was used for the successful modification of the acid-pretreated CNTs onto the oxygen functional groups. Under optimal conditions (0.1 M PBS; pH = 6.5; scan rate 20 mV s^−1^), the electrocatalytic response of the AgNPs/CNTs/GCE sensor showed two prominent and well-separated oxidation DPV throughs, indicating the applicability of the sensor for discrimination of phenolic compounds. Moreover, the amperometric response (examined under the same conditions) responded linearly in the micromolar linear range for both analytes, and with a low LOD value of 0.01 μM for both target species, respectively. Another electrochemical sensor based on AgNPs/MWCNT-coated GCE, this time for the simultaneous detection of four phenolic compounds, was proposed by Athie Goulart and coworkers [152]. In contrast to previous reports [66,80], in this work, AgNPs were firstly synthesized via borohydride reduction, and electrodeposition (10 scans in a potential window from −0.2 V to +0.4 V) on the nanotube walls was performed from as-prepared AgNP solution [152]. The electrocatalytic effect of the AgNPs/MWCNT/GCE sensor was evaluated using a ferro/ferricyanide redox couple, showing a noticeable increase in current (CV analysis), as well as a decrease in *R*_ct_ value and a double amplification in the charge transfer rate (EIS measurements). The SWV technique was used for the simultaneous determination of phenol (+0.83 V), HQ (+0.30 V), CC (+0.40 V), and BPA (+0.74 V) against a sat. Ag/AgCl electrode in local tap water samples without pretreatment, with good sensor reproducibility expressed as the relative standard deviation (RSD) of 2.3%, 3.8%, 2.4%, and 0.8% for each analyte, respectively. 

*Hydroquinone (1,4-benzenediol, HQ) and catechol (1,2-benzenediol, CC)* are two positional isomers of phenolic compounds found as environmental contaminants in many industries (cosmetics, textiles, chemicals, pharmaceuticals, etc.). Two different fabrication designs have been proposed for the simultaneous electrochemical detection of HQ and CC using PANI doped with phthalocyanine-stabilized AgNPs [153] and AgNPs-decorated magnetic Fe_3_O_4_-rGO composite [87], both layered over GCE surface. In the first report [153], a novel hybrid architecture, combining semiconducting and catalytic properties of a macrocyclic molecule (tetraamino cobalt phthalocyanine, TACoPc), with (highly) conducing AgNPs and PANI, showed well-defined CV redox peaks of HQ (+0.68 V) and CC (+0.76 V) vs. standard hydrogen electrode (SHE). Electrochemical detection of analytes (in 0.1 M H_2_SO_4_ primary electrolyte solution) provides a linear micromolar concentration range (DPV), good stability (3.3% and 2.31% changes from the 1st to the 200th cycle for the oxidation of HQ and CC, respectively), and good selectivity to 100-fold concentration of inorganic interferents (Na^+^, Ca^2+^, K^+^, Mg^2+^, Cl^−^, nitrates). Implementation of magnetic nanoparticles (Fe_3_O_4_), presented for the first time in this review as a strategy for sensor fabrication, in combination with the rGO platform and AgNPs provides not only electrochemical sensors, but also photocatalytic self-cleaning ability (degradation of methylene blue and methyl orange as industrial dyes), leading to novel properties [87]. The excellent electrocatalytic capability was demonstrated by DPV analysis, which showed lower LOD values for both analytes compared to the previously presented voltammetric sensor [153]. 

*Nitrophenols (o-NP; p-NP; 4-NP)* are not released from natural sources, but are manufactured and used as a precursors and/or intermediates in the synthesis of nitro dyes and analgesic drugs (acetophenenetidine and paracetamol). A simple voltammetric platform based on uncapped silver nanoclusters (AgNCS) decorated GCE for direct oxidation of 4-nitrophenol was proposed for the first time by the Maduraiveeran group [154]. The AgNCS/GCE sensor demonstrated significantly improved LSV current output and less positive anodic peak potential shift (compared to bare GCE), with direct electrocatalytic oxidation of 4-NP at the potential of +0.63 V/sat.Ag/AgCl and low onset potential (+0.5 V). After the 100th measurement, no significant change in the LSV current peak and the peak potential shift occurred, confirming excellent sensor stability. The cubic-shaped AgNPs supported on rGO platform were used as a GCE modifier for the selective detection of hazardous 4-NP [155]. Activity and stability of the catalyst were confirmed by a chronoamperometry study performed at potentials of −0.3 V and −0.6 V, respectively, in 0.1 M PBS by successive addition of 5 μL of 4-NP. The Ag-rGO/GCE sensor displayed excellent reproducibility (RSD value of 2.87% after five consecutive measurements) and stability (difference in reduction of 13.38% compared to the initial value after 15 days of storage under ambient conditions). Laghrib et al. [156] also used borohydride as Ag^+^ reducing agent to prepare a graphite carbon electrode (GrCE) sensor modified with chitosan-stabilized AgNPs for selective reduction of *p*-nitrophenol. Preliminary CV characterization, conducted within a potential window from 0.4 V to −0.9 V, revealed a pronounced reduction peak at the potential of −0.750 V, which was attributed to the irreversible reduction of the –NO_2_ group to the corresponding hydroxylamine species. The electrochemical behavior of the fabricated *β*-1,4-p-DGA-AgNPs/GrCE sensor was studied in detail (Figure 8a). The effect of scan rate on peak potential and peak current as well as the effect of pH were investigated via CV technique; the results highlighted that the irreversible reduction mechanism involves the exchange of four electrons and four protons, and that the peak potential (*E*_p_) depends linearly on the pH of the medium. Chronoamperometry was employed to study the catalytic reduction of 4-NP. The catalytic rate constant (*K*_cat_) of 1.69 × 10^–4^ mol^−1^ L s^−1^ demonstrated the applicability of the prepared sensor to detect analyte with low detection limit (0.6 μM). The synthesis of AgNPs decorated on tannic acid (TA) covered magnetite (Fe_3_O_4_) has been shown as an outstanding strategy for the development of a highly selective and sensitive 4-NP sensing platform [157]. TA@Fe_3_O_4_-AgNPs/GCE sensor displayed high cathodic current response at a very low overvoltage (−0.39 V) to detect the analyte. Moreover, TA@Fe_3_O_4_-AgNPs nanohybrid showed superior catalytic reduction activity for the target molecule, compared to the sensing platform without nanosilver material. Under optimized conditions (0.05 M PBS; pH = 7.0; scan rate 50 mV s^−1^), this voltammetric sensor showed a broad linear range and a very low LOD value of 33 nM. Hwa et al. [158] prepared SPCE modified with rGO-HNT-AgNPs ternary composite material for selective voltammetric detection of 4-NP in real water samples. Compared to the SPCE modified with binary heterostructures (HNT/AgNPs and rGO/AgNPs), SPCE modified with ternary hybrid structure displayed 2-fold higher catalytic activity. Electrochemical detection of the analyte is based on nitro-groups reduction strategy, with kinetics of the reaction being diffusion controlled mechanism (linear dependence of maximum peak current vs. the sqare root of the scan rate) and pH-dependent process (proton transfer from phenolic hydroxyl groups to produce quinines). Under optimized conditions (DPV resposes in 0.05 M PBS, pH = 7.0; pulse width: 0.05 s; pulse period: 0.2 s; quiet time: 2 s; scan rate 50 mV s^−1^) the prepared sensor exhibited an extensive working range, with practical applicability evaluated in the presence of the analyte (standard addition method) in drinking water, tap water, and industrial water (RSD values of ± 3.17%). SPCE consisted of rGO-FeCo_2_O_4_/curcumin-stabilised, AgNPs-coated WE (*d* = 4 mm), carbon CE, and Ag pseudo-RE, and was used as a voltammetric sensor for simultaneous quantification of 4-NP and hydrazine [159]. Mixed binary transition metal oxides, such as magnetic spinel FeCo_2_O_4_ nanosheets, provide an eco-friendly material of high structural stability and exquisite electrochemical performance. Moreover, the synergistic combination of redox-active Fe and Co sites, together with the electric double-layer rGO material contributed to the facilitated charge transfer, specific capacitance, as well as stability of the sensing electrode. Under optimal conditions (i.e., working potentials of −0.75 V for 4-NP and +0.15 V for hydrazine vs. RE), this ternary composite sensing platform enables simultaneous DPV quantification of target molecules in real samples with satisfied repeatability (RSD of 2.7% and 2.4%, respectively). A ternary nanocomposite modified SPCE was used as a voltammetric sensor for the detection of 4-NP in domestic sewage and underground water matrices in the nanomolar range [160]. An ultrasound-assisted, one-pot synthetic procedure in a batch was introduced to prepare a ternary hybrid nanocomposite by mixing chitosan-modified carbon nanofibers (CS-CNFs) and Ag-doped spinel Co_3_O_4_ nanoflowers. The use of biodegradable and renewable natural polymer improved the hydrophilic nature of the composite material, and the combination of Ag-Co_3_O_4_ enhanced the electrocatalytic property of the sensor to reduce the analyte. This was successfully confirmed with CV, and the DPV analysis performed with the modified SPCE revealed its outstanding electrocatalytic activity (the cathodic and anodic peak potential difference being 29.45 mV, which is ideal for a reversible redox reaction involving two protons and two electrons, respectively), resulting in a LOD value of 0.4 nM. This is also the lowest 4-NP LOD value obtained for any of electrochemical sensors presented in this review. Faisal and coworkers [161] developed Ag-decorated chitosan/SrSnO_3_NC as a suitable sensing platform for the detection of 2,6-dinitrophenol (2,6-DNP) in groundwater, seawater, and tap water samples. To fabricate a functional voltammetric sensor, the nanocomposite was attached to the GCE surface via a PEDOT:PSS linker. By using the irregular perovskite SrSnO_3_ nanocrystal structure in the sensor design, the active sensor surface area was increased (39.7 m^2^/g as determined by nitrogen adsorption/desorption isotherm, i.e., BET analysis), resulting in improved electrocatalytic performance (Figure 8b). DPV was utilized to demonstrate the sensor’s linear micromolar concentration range, achieving a LOD value of 0.18 μM. The sensor provided long-term stability and good reproducibility.

*Bisphenol A* (*4,4′-(propane-2,2-diyl)diphenol*; BPA) is commonly used as a monomer for the synthesis of epoxy resins, and as a food packaging material. In addition, BPA has been classified as an endocrine disrupting chemical (EDS), which can interfere with endogenous estrogen leading to serious health problems (see also Section 3.6). Li et al. [45] proposed a GCE modified with composite material made of rGO, nanosilver, and poly-L-lysine for selective detection of BPA in drinking water and investigated the electrochemical behavior of BPA by CV, EIS, and DPV analysis. Electropolymerization of L-lysine on the surface of rGO-Ag/GCE was done in PBS (pH = 9.0) containing 10 mM L-lysine with CV sweeps in the potential window from −1.0 V to +1.2 V in 10 cycles to improve the electronic properties of the WE. Quantitative determination of BPA under optimal conditions (accumulation potential of −0.4 V and 200 s accumulation time) was performed using the DPV method because of its high sensitivity, rapid detection, and low detection limit. A combination of green synthesized AgNPs (electrocatalytic efficiency) and an rGO platform (large surface area and mechanical stability) layered over ITO substrate served as a voltammetric sensor for the detection of BPA in well water, bottled water, and milk [67]. According to the results obtained from the PBS (pH = 8.0) containing a redox probe in the potential range of −0.8 V to +0.8 V, the peak currents increased notably and the *R*_ct_ value decreased sharply for the AgNPs/rGO/ITO electrode compared to the AgNPs/ITO and rGO/ITO surfaces, indicating increased electron transfer at the electrode/electrolyte interface. The current of DPV responses corresponded to the range of BPA concentrations between 1.9 × 10^−10^ μM and 0.820 μM, with a correlation coefficient of 0.985 and LOD of 0.14 μM.

**Table 3 sensors-23-03692-t003:** Survey of the reviewed silver nanomaterial-based electrochemical sensors for detection of phenolic compounds in water matrices.

Analyte	Sample	Sensor Design/ Detection Method	Linear Range	AgNP Synthetic Approach	LOD	Ref.
Phenol	Tap water, mineral water	AgNPs-M/GCE/DPV	0.8–20 μM	Green synthesis	0.42 μM	[66]
Phenol *o*-cresol	Shale gas wastewater	AgNPs/CNTs/GCE/Amp	10–160 μM 10–200 μM	Electrodeposition	0.01 μM 0.01 μM	[80]
Phenol HQ CC BSA	Tap water	AgNPs/MWCNT/GCE/SWV	2.4–152 μM 2.5–260 μM 20–260 μM 5.0–152 μM	Borohydride reduction	3.1 μM 1.2 μM 1.6 μM 2.4 μM	[152]
HQ CC	Tap water	TACoPc/PANI/AgNPs/GCE/DPV	10–100 μM 10–100 μM	Chemical reduction	0.60 μM 0.46 μM	[153]
HQ CC	River water, tap water, rain water	Ag-rGO-Fe_3_O_4_/GCE/DPV	10–50 μM 10–50 μM	Chemical reduction	37.5 nM 335.4 nM	[87]
4-NP	Aqueous solution	AgNCS/GCE/LSV	0.1–0.6 mM	Borohydride reduction	–	[154]
4-NP	Aqueous solution	Ag-rGO/GCE/Amp	2–150 mM	Borohydride reduction	–	[155]
4-NP	Wastewater, river water	*β*-1,4-*p*-DGA-AgNPs/GrCE/DPV	1–100 μM	Borohydride reduction	0.6 μM	[156]
4-NP	Tap water, drinking water, river water	TA@Fe_3_O_4_-AgNPs/GCE/DPV	0.1–680.1 μM	Hydrothermal	33.0 nM	[157]
4-NP	Drinking water, tap water, industrial water, river water	rGO/HNT/AgNPs/SPCE/DPV	0.1–363.9 μM	Chemical reduction	48.6 nM	[158]
4-NP N_2_H_4_	River water, industrial water	CM-AgNPs/Gr-FeCo_2_O_4_/SPCE/ DPV	2.5–1200 μM	Green synthesis	18.0 nM 23.0 nM	[159]
4-NP	Sewage, underground water	Ag-Co_3_O_4_NFs/CS-CNFs/SCPE/ DPV	0.06–18.93 μM	UV-assisted chemical reduction	0.4 nM	[160]
2,6-DNP	Underground water, sea water, tap water	Ag-Chitosan/SrSnO_3_NC/GCE/ DPV	1.5–13.5 μM	Chemical reduction	0.18 μM	[161]
BPA	Drinking water	RGO-Ag/PLL/GCE/DPV	1.0–80 μM	Hydrazine hydrate	0.54 μM	[45]
BPA	Bottled water, well water, milk	AgNPs-rGO/ITO/DPV	19 nM–0.820 μM	Green synthesis	0.14 μM	[67]

AgNPs-M—*Melia*-coated AgNPs; GCE—glassy carbon electrode; DPV—dynamic pulsed voltammetry; CNT—carbon nanotubes; Amp—amperometry; MWCNT—multiwalled carbon nanotube; SWV—square wave voltammetry; TACoPc—cobalt phthalocyanine; PANI—polyaniline; rGO—reduced graphene oxide; AgNCS—silver nanoclusters; LSV—linear sweep voltammetry; β-1,4-*p*-DGA—poly-D-glucosamine; GrCE—graphite carbon electrode; TA—tanic acid; HNT—halloysite nanotube; SPCE—screen-printed carbon electrode; CM-AgNPs—curcumin-stabilized AgNPs; NFs—nanoflowers; CS-CNFs—chitosan-cabon nanofibers; NC—nanocomposites; PLL—poly-L-lysine; ITO—indium tin oxide.

### 3.5. Electrochemical Sensors for Pharmaceuticals

The surging growth of the pharmaceutical industry has inevitably led to an increase in the misuse and release of pharmaceuticals into the aquatic environment. The main classes of pharmaceuticals found in environmental samples and wastewater include antibiotics, antipyretics, and anti-inflammatories. Due to their toxicity and accumulation in living organisms, their presence poses a serious environmental problem. Therefore, there is a need for accurate, sensitive, portable, and cost-effective technologies to monitor fresh water or wastewater sources for pharmaceutical contaminants [4,137,162].

*Paracetamol* (*N*-acetyl-*p*-aminophenol or acetaminophen, AP) is a potent antipyretic used worldwide mainly for fever reduction and pain relief. High doses and long-term use of acetaminophen can lead to liver disease and nephrotoxicity [5]. Electrochemical methods have proven to be simple, rapid, and accurate methods for the quantitative detection of acetaminophen in clinical diagnosis and quality of acetaminophen-based drugs. An extract of *Araucaria Angustifolia* was used for the first time as a simple and environmentally benign procedure for fabrication of AgNPs [68]. Since the nature and concentration of reducing and stabilizing biomolecules are not the same in every part of the plant, the synthesis parameters were carefully studied and strictly optimized. It was found that AgNPs optimal sensing performance were synthesized at slightly elevated temperature (45 °C), using a 0.50 mmol L^−1^ solution of silver(I) nitrate and an extract:water ratio of 1:0. Transmission electron microscopy (TEM) images of the biosynthesized material revealed almost spherical particles with an average diameter of 91.0 ± 0.5 nm. The as-prepared AgNPs and exfoliated graphene nanoplatelets were deposited on the surface of GCE to fabricate a voltammetric sensor for the detection of paracetamol. The sensor showed excellent analytical features in terms of good repeatability and reproducibility (1.8% and 4.0%, respectively), linear SWV response to the analyte in the micromolar concentration range, and a low detection limit of 8.50 × 10^–8^ M. GCE modified with carbon black (CB); AgNPs and PEDOT:PSS was used for the detection of paracetamol and levofloxacin in river samples [163]. Spherical silver nanoparticles with the median diameter of 10.6 nm were prepared by borohydride reduction. A dispersion of AgNPs (250 μL), CB (1.0 mg), conductive polymer (10 μL), and ultrapure water (740 μL) was drop casted onto the electrode surface. Electrochemical determination by SWV (0.1 M PBS, pH = 6.0) gave a linear concentration range from 0.62 μM to 1.7 μM, with a LOD value of 0.012 μM. The AgNPs-CB-PEDOT:PSS/GCE sensor showed good stability, reproducibility, and repeatability, and no interference occurred in the electrochemical detection of the analyte in the presence of glucose, caffeine, and urea molecules. rGO decorated with a suspension of Ag-Pd bimetallic nanoparticles (1 mg mL^−1^) and 1% chitosan solution (CS) was dropped onto the GCE surface to fabricate a voltametric sensor for the determination of acetaminophen [164]. The CS/Ag-Pd@rGO/GCE sensor showed high stability (96.3% of the initial value after 20 days storage at 4 °C) and good selectivity (1000-fold concentration of NaCl, KCl, CuSO_4_, and CaCl_2_ did not affect the paracetamol determination). A nanohybrid structure of AgNPs integrated into a porous 3D metal–organic framework (ZIF-67) and deposited on GCE demonstrated enhanced electrocatalytic activity toward dopamine (DA) and acetaminophen [165]. Such a specific morphology (dodecahedral crystallite imaged by TEM) enlarges the active composite site and increases the specific surface area. The DVP responses of DA and AP on Ag-ZIF-67p-modified GCE (0.10 M PBS, pH = 7.0) revealed two separated anodic peaks with good linear relationships between anodic peak currents and DA/AP concentrations. The exquisite electrochemical performance of the prepared sensing surface was ascribed to the synergistic interaction between ZIF-67 and AgNPs nanomaterials. A schematic presentation of the sensor development is presented in Figure 9.

The supramolecular recognition performance of cationic pillar[5]arene (CP5) towards AP was used as a strategy to develop an electrochemical sensor [166]. Moreover, CP5 has also been used as an efficient ligand for borohydride-reduced AgNPs. The macrolytic host–guest recognition study was investigated by fluorescence quenching (excitation wavelength of 289 nm); the calculated CP5/AP binding constant of (3.37 ± 0.26) × 10^4^ M^−1^ revealed a strong selectivity of the recognition ability of CP5 toward acetaminophen. CP5-AgNPs and acid sulfated cellulose nanocrystals (CNCs) were mixed in a volume ratio of 2:1, and 6 μL of the obtained suspension was drop casted onto the surface of the GCE. Quantitative AP detection on the CNCs@CP5–AgNPs/GCE sensor was provided using DPV (sweep range 0.2–0.8 V; pulse width 0.05 s; sampling width 0.02 s). Compared to the previously described sensors, a wider linear micromolar concentration range and the lowest AP nanomolar detection limit were achieved.

*Dopamine* is a natural monoamine synthesized in plants and animals that functions as a key neurotransmitter [167]. In addition, dopamine is also used in pharmacology. A silver-molybdenum disulfide composite material (Ag@MoS_2_), drop casted onto the GCE surface, was employed for highly selective aqueous DA detection in the presence of uric acid (UA) and ascorbic acid (AA) as interferences [168]. To prevent rapid reduction of silver(I), i.e., to precisely control the growth rate of Ag nuclei, the green synthesis approach was conducted in alkaline media by forming [Ag(NH_3_)_2_OH] complex as intermediate. High resolution (HR)-TEM images showed the growth of ultrafine AgNP spheres with narrow size distribution on the MoS_2_ nanosheets with an average diameter of 0.8 × 0.2 μM. Compared to the bare GCE and MoS_2_-GCE sensors, the as-prepared sensor showed increased anodic and cathodic peak currents (CV experiments) and a considerable decline in the Nyquist semicircle diameter at higher frequencies (EIS analysis), indicating rapid electron transfer at the interface due to the presence of AgNP in the hybrid architecture. Under optimal conditions (N_2_-saturated 0.1 M PBS; pH = 7.2), sharp DPV responses were achieved, with oxidation current maxima linearly proportional to DA concentration in the range of 1–500 μM. After 10 days of storage in PBS at 4 °C, excellent stability of 95% of the original current response was obtained. In order to ameliorate the electrocatalytic activity of functional silver nanomaterial towards DA, a platinum-silver graphene (Pt-Ag/Gr) bimetallic nanocomposite was introduced in voltammetric sensor design [169]. Electrochemical oxidation of DA on the prepared sensor is a surface-adsorption-controlled reaction (redox current trough increases with the scan rate), which involves the transfer of two protons (pH-dependance) and two electrons (effect of scan rate). Therefore, quantitative analysis of DA on the Pt-Ag/Gr/GCE sensor was investigated in 0.1 M PBS (pH = 6.5), applying the DPV method. Although this sensor exhibited shorter dynamic range than the previously presented DA voltammetric sensors [165,168], employment of bimetallic structure in conjunction with graphene-supporting material resulted in lower LOD value, i.e., better sensor sensitivity [169]. Table 4 summarizes nanosilver-based electrochemical sensors for detection of phenolic compounds.

### 3.6. Electrochemical Sensors for Nitroaromatics

Due to their high solubility in water and low vapor pressure, nitroaromatic compounds (e.g., nitrobenzene, nitrotoluene, nitroaniline, nitroaromatic drugs) have been found to be significant environmental pollutants and have been associated with toxicity, carcinogenesis, and mutagenesis.

Pandiyarajan et al. [170] prepared AgNPs-supported, graphitic-like C_3_N_4_ nanocomposite via chemical reduction to detect *nitrobenzene* (NB) in aqueous samples. The AgNPs were stabilized with *N*-[3-(trimethoxysilyl)propyl]ethylenediamine (EDAS) via an amine group to improve the anchoring of AgNPs onto the graphitic nitride sheets and prevent the aggregation of the particles. The nanocomposite has an electrochemical detection function for analyte, with SWV reduction peaks showing a linear dependence on NB concentration in broad concentration range. The sensitivity of the EDAS/(g-C_3_N_4_-Ag) nanocomposite was found to be 0.594 A M^−1^ cm^−2^, with a LOD value of 2 μM. Bimetallic gold–silver alloy nanodots (AuAgNDs) with a size of less than 3 nm, encapsulated in a silicate sol–gel (SSG) matrix functionalized with amine groups, were found to be a more environmentally friendly electrocatalyst for the monitoring of NB [83]. The AuAg alloy expressed a characteristic SPR band between the AuNPs and AgNPs, and a unique electrocatalytic behavior compared to monometallic-coated GCE sensors. Furthermore, the synergistic effect of the small AuAg alloy size, combined with the preconcentration of the analyte due to SSG porosity, resulted in a threefold improvement in NB reduction peak current. A linear dependence of the SWV reduction peak current was noticed with a correlation coefficient of *R*^2^ = 0.998, and a sensitivity of 0.045 μA/μM towards nitrobenzene reduction. Shivakumar and coworkers [171] devised a sensitive voltammetric sensor for the detection of NB on *Eucalyptus bark* synthesized AgNPs immobilized on the GCE surface. It was found that the modified electrode made an effective electrocatalytic contribution to reduce NB and to diminish the analyte overpotential; therefore, the peak cathodic current increased compared to the bare GCE. The DPV responses obtained were linear at different concentrations of the analyte (5–45 μM range) in N_2_-saturated 0.1 M PBS, with a sensitivity of 2.262 μA μM^−1^ cm^−2^ and a calculated LOD value of 0.027 μM (*S*/*N* = 3). The modified electrode was successfully used as a sensor for detecting NB in real water matrices, with recoveries between 98.42% and 102.18% (tap water), and 97.92% and 99.84% (lake water), respectively. A wider linear range and a lower LOD value for the electrocatalytic reduction of NB was achieved with *Camellia japonica* leaf-extract-synthesized AgNP-coated GCE [172]. The typical amperometic current–time behavior (applied potential of −0.42 V; 0.05 M PBS; pH = 7.0) was obtained with the continuous nitrobenzene addition under constant stirring; the reduction peak current changed linearly with respect to the concentration of the analyte in the lower and upper concentration ranges. This behavior can be attributed to the adsorption of nitrobenzene reduction products on the surface of the green catalyst, changing the reduction kinetics. However, the importance of this sensor lies in its dual role, as it is also suitable for photocatalytic degradation of dye Eosin-Y. The development of a sustainable method to remediate Hg(II) as Hg(0) using graphene quantum dots coated carbon cloth (CC/GQDs) and utilization of Hg(0) and AgNPs/CC/GQDs composite as a sensitive electrochemical sensor for nitrobenzene was reported [173]. This sensing platform not only reduced the overpotential for analyte reduction by more than 100 mV, but also exhibited a 4-fold increase in the oxidation current compared to the bare electrode. The NB DPV reduction current (0.2 M PBS, pH = 7.0) increases linearly with analyte concentration within a wide linear range (*R*^2^ = 0.9943), rendering the lowest nitrobenzene LOD value of 30 pM. The applicability of the presented sensor was tested in spiked real matrices (20 and 10 ppm of Hg(II)), and the total Hg(II) removal efficiencies of 24% (lake water), 20% (tap water), 48% (river water), and 11% (sea water), respectively, were obtained.

A robust chronoamperometric sensor for the detection of *p*-*nitrotoluene* (*p*-NT) based on a borohydride-synthesized, spherical, AgNPs-coated Au disc electrode was proposed by Rani et al. [174]. The as-prepared Ag/AuE sensor, coated with a Nafion layer, showed good electroanalytical response in the presence of *p*-NT in tap water samples. The operation of the sensor at a constant potential of +0.050 V displayed fast response time, good sensitivity (6.36 μA/μM cm^2^), and a sensibility toward the targeted molecule in the presence of the interferents. The response to the analyte was found to be linear in the concentrations ranging from 0.01 μM to 10.0 μM, with 0.092 μM LOD. Laghrib et al. [175] fabricated a sensitive voltammetric sensor for the detection of *p*-*nitroaniline* (*p*-NA) on chitosan-encapsulated AgNPs CPE. It was found that the modified electrode made an effective electrocatalytic contribution to the reduction of *p*-NA, with a large decrease in the *R*_ct_ value (obtained from the Nyquist semi-circle curve) compared with the bare CPE. The CS-SNPs/CPE sensor exhibited a wide linear concentration range, with 7 nM LOD for quantification of *p*-NA. The modified electrode was characterized by good selectivity (4-nitrophenol and 2-nitroaniline interferences), reproducibility (RSD = 4.31% for eight consecutive measurements), and applicability in conventional aqueous samples.

*Nitrofurantoin* (NFT) is a commonly used antibiotic drug for the treatment of urinary tract infections, whose overdosed side effects include abdominal pain, headache, depression, diarrhea, and dizziness. Silver-capped selenium particles were used to fabricate an electrochemical sensor for the simultaneous detection of NFT in aqueous samples [176]. The microrod-shaped morphology of selenium particles (average size of 257.33 nm; obtained from TEM images), decorated with rocky-shaped nanosilver material (average cluster size of 145.32 nm), served as an active site for the accumulation of target analytes. By varying the scan rates using 5 mM ferri/ferrocyanide in 0.1 M KCl, a large electrochemically active sensor surface area (0.2662 cm^2^) was detected. In addition, by varying the scan rates with 50 μM NFT in PB buffer (pH = 7.0), the prepared Ag/Se/GCE sensor also exhibited high electrocatalytic properties (charge transfer coefficient of 0.8884) and kinetic parameters (heterogeneous standard rate constant of 7.4958) towards NFT reduction. The exquisite analytical parameters of the DPV sensor were reflected in lower NFT detection (9.86 μM) compared to the conventional HPLC technique (9.95 μM) in tap water samples. Kokab et al. [177] reported the development of an ultrahigh sensitive voltammetric sensing platform equipped with silver nanoparticles-functionalized carbon nanotube (COOH-CNTs/Ag/NH_2_-CNTs) for the detection of amlodipine (AM) and atorvastatin (AT) drugs in tap and drinking water. The sandwich nanocomposite material was immobilized, characterized morphologically (SEM, TEM) and structurally (X-ray diffraction, Fourier transform infrared spectroscopy), and then applied to the GCE. Compared with the working area of the bare GCE (0.02 cm^2^); a sevenfold magnification was obtained for the electrode decorated with nanocomposite (0.14 cm^2^). Likewise, the Nyquist plots revealed the passive electrochemical diffusion process (*R*_ct_ value of 6379 Ω), while a much lower charge transfer resistance value (2.0 × 10^–5^ Ω) was obtained for the modified GCE, pointing on faster charge transduction at the electrode/solution interface. The SWASV method was established for the successful detection of AM and AT drugs and was also supported by the density functional theory (DFT) approach. The preconcentration step enables enhanced selective adsorption of target molecules at sensing platform through π-π stacking between CNTs and aromatic analyte rings, boosting the electrooxidation of the AM and AT molecules at the AgNPs active sites and enabling their quantification at femtomolar concentration. Table 5 summarizes nanosilver-based voltammetric and amperometric sensors for detection of nitroaromatics.

### 3.7. Electrochemical Sensors for Natural and Synthetic Estrogens

Hormones are key messengers that convey essential information in living organisms and are used to regulate physiological processes [178]. Among them are estrogens, a group of chemically similar steroid hormones found in animals and humans. Both biogenic (estrone (E1), 17-*β*-estradiol (E2), estriol (E3), and estretol (E4)) and synthetic estrogens (dienestrol, 17-α-ethinylestradiol) are released into the environment through pharmaceutical and agricultural industries, making them emerging pollutants found in the aquatic environment and even in drinking water [179]. Because they interfere with normal physiological processes, biogenic and synthetic estrogens are also classified as endocrine disrupting chemicals (EDCs). Prolonged exposure to EDCs through food and drinking water can lead to reproductive disorders and tumors in humans. Therefore, strict monitoring of their presence in food and environmental water specimens using selective and sensitive methods is essential for human health [7]. Table 6 summarizes nanosilver-based electrochemical sensors for detection of natural and synthetic estrogens in water matrices.

Among natural estrogens, *17-β-estradiol* is the main bioactive molecule that enters into the environment through sewage and animal feed production [180]. Graphene-coated silver nanoparticles (GN@Ag) anchored over graphitic carbon nitride (g-C_3_N_4_) have been proposed as an outstanding hybrid nanocomposite material for electroanalytical quantification of E2 [181]. In ultrasound-assisted reflux fabrication approach, the graphene–silver NPs were randomly dispersed over g-C_3_N_4_ sheets (imaged via SEM and TEM, respectively). Such prepared voltammetric GN@Ag/g-C_3_N_4_/GCE sensor exhibited excellent conductivity and enabled rapid electron transfer, showing excellent CV response to the oxidation of E2 as well as linearity over a wide micromolar concentration range. This simple sensing platform was remarkably stable (91% of initial current response obtained after 21 days) and applicable in real water matrices with recoveries between 95% and 104%. A voltammetric sensor based on molecularly imprinted polymer (MIP) and 2-mercaptobenzoxazole (2-MBO)-capped AgNP composite material was presented as a novel sensing platform for the highly sensitive and selective determination of E2 in real water samples [182]. First, the GCE was modified with nanosilver (electrodeposition from 0.5 mM AgNO_3_ with addition of a capping agent in 0.1 M PBS under the potential range of −0.7 V to 1.4 V for 7 cycles). Second, electrochemical polymerization of poly(*p*-aminophenol, *p*-APh) was performed in the presence of the E2 molecule as an MIP template (10 mM p-APh, 1 mM E2, and PBS-acetone mixture, pH = 3.0, scan rate of 100 m V s^−1^ controlled between −0.40 V and +0.95 V for 10 cycles). The number of potential cycles during polymerization (5, 10, 15, or 20), polymerization scan rates (50, 100, 150, and 200 mV s^−1^), effects of pH, and incubation time were studied: the number of scan cycles of 10 proved to be optimal for sensor preparation; the scan rate of 100 mV s^−1^ gave the highest peak current; neutral pH was suitable for E2 detection; and 15 min was the required incubation time. SWV measurements (*n* = 3) showed that the 17β-estradiol MIP sensor responded to the increasing E2 concentrations from 10 pM to 100 nM, and had a LOD value of 1.86 pM. An electrochemical MIPs-nanocomposite sensor, this time made of ternary poly-imidazole (PImi)-graphene oxide (GO)-AgNPs hybrid material for the selective detection of E2 in river water, was fabricated by the same authors as in the previously published work [183]. The nanosilver material serves as a catalyst and charge carrier, imidazole-based MIP (*p*-type-electron acceptor) provides specific binding cavities for the analyte, and the role of GO (*n*-type-electron donor) in the composite material is to provide additional functional units to bind the template. The imprinted polymer-modified GCE (optimized for Imi to E2 molar ratio, scan rate, scan cycle, pH range, template removal, and the electrode incubation) exhibited relatively small SWV responses after the E2 incubation of 9 min, suggesting that diffusion of ferry/ferrocyanide redox probe to the sensor surface was blocked due to the large amount of analyte detected. The synergistic effect of the MIPs, GO, and AgNP ternary nanocomposite with enlarged high surface-to-volume ratio showed exceptional analytical sensor features, yielding a femtomolar LOD value.

*Estriol* ((16α,17β)-estra-1,3,5(10)-triene-3,16,17-triol, E3) is an estrogenic hormone used to prevent and treat disorders caused by hormone deficiency, cancer, or urogenital diseases in women [180]. Carbon black nanoball (CNB)-shaped particles (average diameter of ~20 nm), decorated with 5–6 nm sized AgNPs, were used as a GCE modifier layer for the oxidation of E3 hormone [184]. The authors have shown that the electron exchange process at the electrode/solution interface is strongly influenced by the surface chemistry. The GCE modified exclusively with CNBs exhibited slow electron transfer kinetics (*R*_ct_ value of 770.97 Ω) due to the abundant oxygen groups, which decreased significantly to a value of 125.82 Ω as the degree of reduction of Ag(I) onto the CNB surface progressed. Moreover, the estriol oxidation peak potential shifted to lower values when composite material was used, and showed higher current maxima. The proposed reaction sequence of electrochemical oxidation mechanisms, in which the analyte is oxidized first to phenoxy radicals, and second to phenoxonium ions chemically coupled to the ketone form, is consistent with the electron transfer chemical process involving two protons and two electrons. The CNB-AgNP/GCE sensor was reproducible, stable, and robust against major E3 interfering species. A synergistic combination of rGO and AgNPs was used to develop a modified GCE for the successful detection of estriol in tap water samples [185]. By varying the amounts of GO and AgNO_3_ reactants, the authors confirmed that the composition of the electrode material is vital for achieving the superior analytical features of the sensor; it was shown that the best current output was obtained with 20 wt % of the nanosilver precursor in the nanocomposite. To obtain broader linear concentration range, as well as a lower LOD, pretreatment step (applying a potential of −0.80 V for 30 s) was performed for the concentration of estriol molecules near the modified GCE surface. The oxidation mechanism of the analyte, which was elucidated by the combination of electrochemical experiments (CV and DPV) and computational approaches (molecular dynamics simulations), revealed an irreversible oxidation process in which one electron and one proton are transferred between the estriol molecule, WE, and the buffer solution media, respectively. The prepared sensing platform was used for the determination of the estriol hormone in tap water samples (*n* = 3) with recoveries ranging from 94.6% to 108.2%.

A composite of AgNPs- and SWCNTs-modified CPE exhibited attractive properties for the determination of the synthetic estrogen *dienestrol* (4-[4-(4-hydroxyphenly)hexa-2,4-dien-3-yl]phenol; DNL) [84]. In order to find the reaction medium with the most pronounced synthetic estrogen electrooxidation signal, different buffer and acid solutions (pH range 2–12) were tested in a conventional three-electrode system, using CV at scan rate of 50 mV s^−1^. The molecule was found to be more reactive under acidic conditions (BR buffer, pH = 2.0), with irreversible oxidation of the phenolic dienestrol group occurring at the potential around +1.0 V. Moreover, the experimental findings confirmed that the electrochemical–chemical mechanism involves two electrons and two protons, via a stable reaction intermediate. The SWV current signal associated with the formation of quinine derivatives can be used to monitor DNL content in river water, without the need to pretreat the samples and without the accuracy of the sensor being affected by the presence of the natural organic matter (recoveries between 98.8% and 100.2%, respectively). The nanomolar DNL detection level indicates on the feasibility of the proposed sensor for screening synthetic estrogens in water samples.

**Table 6 sensors-23-03692-t006:** Survey of the reviewed silver nanomaterial-based electrochemical sensors for analysis of synthetic and natural estrogens in environmental water samples.

Analyte	Sample	Sensor Design/ Detection Method	Linear Range	AgNP Synthetic Approach	LOD	Ref.
E2	Environmental water	GN@Ag/g-C_3_N_4_/GCE/Amp	0.005–8.0 μM	-	0.002 μM	[181]
E2	Tap water, bottled water	MIPAPh-AgNP/GCE/SWV	10 pM–100 nM	Electrodeposition	1.86 pM	[182]
E2	River water	MIP-GO-AgNP/GCE/SWV	10 fM–250 nM	Electrodeposition	3.01 fM	[183]
E3	Creek water	CNB-AgNP/GCE/DPV	0.2–0.3 μM	Polyol method	0.16 μM	[184]
E3	Tap water, urine	rGO-AgNPs/GCE/DPV	–	Borohydride reduction	21.0 nM	[185]
DNL	River water	AgNP/SWCNT-CPE/SWV	29–543 μg L^−1^	Chemical reduction	43.7 nM	[84]

GN@Ag/g-C_3_N_4_—graphene nanosheets-silver-graphitic carbon nitride; GCE—glassy carbon electrode; Amp—amperometry; MIPAPh—molecularly imprinted poly(*p*-aminophenol); SWV—square wave voltammetry; GO—graphene oxide; CNB—carbon black nanoballs; DPV—differential pulse voltammetry; rGO—reduced graphene oxide; SWCNT—single-wall carbon nanotubes; CPE—carbon paste electrode.

## 4. Discussion

This review of voltammetric and amperometric sensors for the detection of emerging inorganic and organic pollutants in aquatic environments highlights recent advances, associated challenges, and future directions. A summary of key detecting approaches for specific environmental pollutant classes during the selected timeframe (2017–2022) is presented in Table 1, Table 2, Table 3, Table 4, Table 5 and Table 6, including the sensor design and the electroanalytical detection method used, the synthesis method for the preparation of the functional nanosilver material, and the analytical parameters of the developed sensing platform (linear range and LOD value). The schemes of the electrochemical platforms along with the corresponding detection mechanism of the selected sensing architectures are also provided for each class of hazardous substances (Figure 1, Figure 2, Figure 3, Figure 4, Figure 5, Figure 6, Figure 7, Figure 8 and Figure 9). A detailed discussion will highlight the importance of chemically modified electrodes. The relationship between the simplicity of reduction methods in the batch for the fabrication of AgNPs as a highly functional sensing probe and electrocatalytic active site; the mechanism of AgNPs-based voltammetric and amperometric sensing platforms, and the functional design of electrochemical devices for selective detection of aquatic pollutants, is provided below. The main challenges are observed and ways to overcome them are suggested. The discussion will further extended to cover the future prospects of this appealing research area.

### 4.1. Electrochemical Sensor Technology

Electrochemical sensor technology has become an important aspect of modern analytical chemistry, and great efforts are being made to develop novel and cost-effective sensing platforms that provide a fast, accurate, and repeatable response to the analyte of interest. Compared to ubiquitous colorimetric sensors, the main advantage of electrochemical detection is the ability to measure in turbid samples [18]. Moreover, working electrodes can also be made in a planar and/or flexible mode [67,113,143,148], i.e., they can be easily miniaturized and integrated into more hierarchical devices [186]. Electrochemical sensors can be used in various fields, including pharmaceuticals, medicine, chemistry, synthesis, materials engineering, and biotechnology [137,187,188]. In addition, electrochemical sensors based on disc (3D) or planar (2D) electrodes decorated with AgNPs have proven to be simple but reliable analytical tools for the rapid and selective detection of emerging water pollutants (i.e., heavy metal ions, nitrogen-containing inorganic species, phenolic compounds, nitroaromatics, pharmaceuticals, and natural and synthetic hormones), which is highlighted in this review.

### 4.2. Wet Chemical Synthesis of AgNPs for Sensing Applications

For the fabrication of highly selective and sensitive voltammetric and amperometric electrochemical sensors, the usage of silver nanomaterials with precise particle size, shape, crystalline facets, and morphology is essential. Hence, controlled synthesis is a paramount for achieving excellent electrocatalytic activity for practical sensing application. AgNPs are usually synthesized using a chemical reduction from soluble silver(I) precursors (mostly AgNO_3_), and many of these synthetic processes are carried out under harsh reaction conditions applying toxic hydrazine [45] or borohydride [113,163,166] reducing agents and/or (volatile) organic solvents [149]. Although there has been more recent focus towards green synthesis [31,38,189], in 64 of the 86 electrochemical sensors presented in this review the reduction of silver(I) to zero-valent form was assisted by electric current or inorganic reducing agents. It is important to emphasize that if the chemical reduction process using strong inorganic agents is precisely controlled, i.e., if no hazardous chemicals are released into the environment during the synthesis [44], such an approach does not necessarily contradict to the principles of green synthesis. Moreover, AgNPs of smaller size and cleaner particle shapes can be produced with strong reducing agents: (i) nanoball-shaped AgNPs (*d* = 5–6 nm) via the polyol process [184], (ii) spherical crystals (mean diameter of 10.6 nm) obtained via borohydride reduction [163], or (iii) globular particles with a diameter of ~20 nm via ethanolamine reduction [135]. On the other hand, biosynthesized AgNPs generally exhibit irregular morphologies, are polydisperse, and often form larger particle agglomerates [67,129,171]. Nevertheless, even strong reducing agents can render AgNPs of various morphologies (quasi-spherical, cubic, twinned structure, and triangles) and larger size distributions when deposited on highly wrinkled and folded graphene nanosheets [155], or when the Ag(I) precursor is previously encapsulated with a massive chelating ligand [156], while a precisely controlled green approach can produce remarkably small particles (*d* = 5 nm) with high catalytic efficiency [168]. AgNPs of precise size and shape can also be formed and fine-tuned using electric current [79,80,119,182]. In electrochemical sensors based on electrodeposited silver nanomaterial [81,148], the addition of silver material was found to play a key role in the trace-level quantification of both trivalent and hexavalent chromium through formation and stabilization of functional bimetallic silver–gold metal oxides [81], and in the formation of dual-region WE for simultaneous detection of inorganic nitrogen-containing species (nitrate/ammonia) through a signal current channel [148].

### 4.3. Electroanalytical Techniques for Characterization of AgNPs-Modified Electrodes and Analyte Quantification

In general, cyclic voltymmetry and impedance measurements are used to study the changes in the interfacial phenomena between the sensor body and solution when different electrode coating materials are used. In other words, CV and the EIS technique are indispensable tools to observe in detail the stepwise evolution of the electrochemical sensor fabrication pathway. To implement both techniques the usage of a ferri/ferrocyanide redox probe is needed, and measurements are usually performed in aqueous 0.1 M KCl [103,133,141,171,176] or in a phosphate buffer solution [159,166]. With the introduction of AgNPs as electrode modifier material, both the increase of current peak and the reduction in peak separation are noticeable with cyclic voltammetry [184]. The AgNPs modification of the bare electrode is also evident in electrochemical impedance spectroscopy, where the value of electron transfer resistance, which corresponds to the semicircle diameter of the Nyquist diagram, is significantly lower compared to the bare electrode (Figure 10). Furthermore, the diffusion-limited process is represented by the linear portion of the Nyquist plot at lower frequencies, while the charge-transfer-limited process is related to the part of the diagram at higher frequencies. In summary, the specific role of the nanosilver material in the exchange of the electron transfer rate between the electroactive species (redox couple) and the modified sensing surface is attributed to the enhancement of the maximum peak current (CV analysis) and the diminution of the impedance (EIS analysis).

Voltammetry and amperometry are particularly attractive techniques for the analysis of complex sample matrices such as various environmental waters (groundwater, lakes, rivers, streams, seawater, etc.) The choice of voltammetric technique depends primarily on the location of the sample and the expected concentration of the analyte. Although CV can be used to determine analyte concentration [65,97,138,148], DPV or SWV methods have been shown to be better for quantitative analysis. SWV is known for its superior sensitivity and measurement speed compared to DPV technique [68]. For greater sensitivity, analyte preconcentration can be performed by adsorptive square wave stripping voltammetry [69,177].

### 4.4. Mechanisms of AgNPs-based Voltammetric and Amperometric Sensors

The detection mechanism of nanosilver-based voltammetric and amperometric sensors for water pollutant monitoring summarized in this review is based on several different mechanisms: (i) AgNPs aggregation; (ii) AgNPs displacement; (iii) AgNPs electronic conductive channels enhancement/inhibition via selective site recognition; and (iv) OR reaction of inorganic/organic water pollutants.

(i)Aggregation of AgNPs triggered by the addition of the analyte is a common optical sensing approach that can be converted to voltammetric [98] and amperometric sensors [101]. This strategy is feasible usually in the form of dual- [98,99] or multisensing [102,106,111,122] platforms, with optical, fluorescent, and electrochemical response towards the detection of HMs. In contrast to the colorimetric assay, electrochemical aggregation of functional silver material results in voltammetric/amperometric signal amplification or inhibition, depending on the chemistry of the nanoparticle stabilizer and the target analyte. For the aggregation-induced sensing mechanism, the role of biogenic synthesized AgNPs is highlighted because biogenic molecules are rich in electron-donor groups that successfully act as analyte-chelating ligands. AgNPs synthesized using fungus *Agaricus bisporus* [98] and the *Mimosa diplotricha* leaf extract [99] were found to be potent sensors towards hazardous mercury, boosting the DPV responses through strong metal–nanosilver stabilizer complexation. Green AgNPs also play a prominent role as multisensor probes for mercury [102], cadmium [106], arsenic [120], and chromium [122].(ii)Electrochemical sensors based on the displacement of functional nanomaterials are considered a novel tool for highly selective and sensitive detection of analytes. In two examples of the reviewed scientific papers, the mechanism of mercury detection was associated with the removal of the surface-bound stabilizer (calixarene moiety), leaving the AgNPs uncapped [101], or with galvanic displacement of silver by mercury [104]. As thermodynamically unstable species, the bare nanoparticles tend to aggregate into larger clusters, or form a silver—mercury amalgam [101]. This leads to signal inhibition, which is visible in the optical mode by a reduction in the SPR band in the UV-Vis spectrum, and in the amperometric mode by a reduction in current output. In the galvanic displacement method, the AgNPs interact with divalent mercury ions in the solution and convert the zero-valent silver into ionic silver species, resulting in loss of stripping signal. Since only small amounts of the analyte are required to displace measurable amounts of silver, this sensor can quantify inorganic mercury in the picomolar range.(iii)Aggravation or enhancement of the electron transfer pathway of AgNPs-modified WE leads to the amplification or decrement of voltammetric performance. Two voltammetric platforms based on MIPs supported by AgNPs-decorated GCE have been developed for highly selective and sensitive detection of natural estrogen in environmental and drinking water with picomolar and femtomolar detection limits [182,183]. The detection of the analyte is catalyzed by the AgNPs (signal amplification or inhibition), while the analyte capture and selective recognition were performed by the MIPs. In the presence of E2, more imprinted cavities of the poly(*p*-aminophenol) (MIP-pAPh) were occupied, resulting in direct amplification of the current response by AgNPs signal amplification [182]. The reverse sensing mechanism was proposed for the AgNP/GO/MIP (poly-imidazole; PImi) functional platform [183]. In addition to the imidazole moiety (*p*-type-electron acceptor), additional functional sites of GO (*n*-type electron donor) were available for analyte binding, which blocked the diffusion of the ferri/ferrocyanide redox probe at the electrode/solution interface, reducing the SWV current responses.(iv)Voltammetric and amperometric sensors, which detect inorganic/organic pollutants based on their OR response, are the most represented electrochemical sensor class. AgNPs increase the active surface area of the bare and/or modified electrode, facilitate electron transfer, and serve as active catalytic sites for oxidation/reduction of the aquatic pollutant. It has been reported that the reaction rate constant (*k*) is proportional to the total effective surface area of the nanosilver material [190]. Therefore, the homogeneity of particle size and shape of AgNPs anchored on sensing platforms is of utmost importance for catalytic efficiency. The adsorption of the analyte (the first step in the sensing mechanism) is more promoted on small particles with larger effective surface area, which facilitates the OR process of electroactive species on the modified electrodes [135]. The interactions between the AgNPs, electrode support and/or other electrode modifiers, and the target analyte, depends on the nature of the surface-active species. On the one hand, stabilizers can be interpreted as a barrier between the AgNPs and the electrode material, controlling the electron transfer kinetics and, thus, affecting the analytical performance of the sensor, especially in terms of achieving lower detection limits [103]. Therefore, clean and uncovered active surface sites of nanoclusters are preferred for catalytic and sensing applications [154]. However, since unprotected AgNPs tend to aggregate, the presence of a stabilizer is extremely important. The catalytic activity of green-capped AgNPs can be altered by the size of the stabilizer (bulky ligands or small molecules) [66,166], and by the adsorption of intermediary molecules (formed during the sensing mechanism) onto the biostabilizers, as was the case in the amperometric quantification of nitrobenzene [172]. The determination of nitrites based on their irreversible electrochemical oxidation via two-electron transfer was reported for various AgNPs-functionalized electrodes, i.e., nanospheres decorated GCE [129], core-shell Au@Ag structure anchored over carboxylated graphene/GCE [86], AgNPs-(r)GO platform [130,131], AgNPs/rGO nanohybrid in conjunction with PPy [133] and PANI [134] conducting polymers, AgNPs/ZnO nanocomposite [136], and AgNPs/HNT/MoS_2_ platform [139]. In contrast, one-electron transfer was demonstrated in nitrite sensing with a nanosilver–polymer composite (chitosan, PEDOT:PSS) [142], and also with an AgNPs-MWCNTs hybrid platform [74]. AgNP/SWCNT/CPE was used to detect DNL by the irreversible oxidation of phenolic groups to (semi)quinone species [84]. The overall chemical–electrochemical mechanism involves a transfer of two protons and two electrons, but the final product depends on the applied potential and the electrode modifier used. Oxidation products of E3, obtained using GCE coated with AgNP/CNB [184] and AgNPs-rGO [185], correspond to ketone derivatives. For ultrasensitive detection of AM and AT drugs, an oxidation mechanism involving rapid electron transfer from the analytes through the COOH-CNTs/Ag/NH_2_-CNTs ternary sandwich architecture to the GCE was proposed [177]. The overall electrochemical oxidation (single SWASV profile) of the electroactive center 1,4-dihydropyridine (AM) occurs through a two-electron/two-proton process, whereas the pyrrole center of AT exhibits two-electron/one-proton transfer. The reversible oxidation of CC and HQ phenolic compounds to *o*- and *p*-benzokinone species via a two-electron/two-proton transfer was mediated by the cobalt active site of the AgNPs/TACoPc/PANI ternary composite using the DPV technique [153] and with AgNPs/Fe_3_O_4_-rGO hybrid material [87]. Quantitative determination of BPA via electrooxidation to 2,2-bis(4-phenylquinone) was successfully performed with an AgNPs-rGO composite on a ITO substrate [67], and with an AgNPs-rGO/PLL-coated GCE [45]. Conversely, several phenolic compounds were detected using the reduction mechanism. The nitro groups of 4-nitrophenol [154,156,158], 2,6-dinitrophenol [161], 4-nitrotoluene [174], nitrofurantoin [176], and nitrobenzene [170,171,172] were irreversibly converted into hydroxylamine or aminophenol groups by an electrochemical mechanism involving four protons and four electrons. Moreover, nitrobenzene was detected on a silicate sol–gel matrix GCE functionalized with AgAu alloy, depending on the electrochemical reduction of the nitro group involving six protons and six electrons [83]. Gold electrode coated with AgNPs [146,149], in conjunction with the SWV technique, proved to be a highly efficient combination for the detection of nitrates in seawater due to their two-electron reduction to nitrites. In addition, nitrate was detected on a Cu(II)-terephthalate decorated SPCE via a Cu^2+^/Cu^+^ redox couple [46]. The adsorption of nitrates is promoted on the Cu(I) active site by copper–oxygen coordination interactions, which is further supported by the charge transferability of MOFs and the enhanced transfer of two electrons to the catalytic AgNP sites.

### 4.5. Design of AgNPs-based Voltammetric and Amperometric Sensors for Detection of Aqueous Pollutants

The choice of bare electrode substrate (supporting material) is of paramount importance in manufacturing process of electrochemical sensors. The common denominator in the use of pure metal electrodes (Au or Pt), or bare carbon-based electrodes (GCE, GPE, PGE, CPE, SP(C)E, etc.), is the difficult electron transfer between the electroactive analyte molecule and the sensing electrode materials. This leads to low selectivity towards the analyte (the redox reaction is possible at potentials substantially higher than thermodynamic potentials), and inability to distinguish the target molecule from interfering species in real matrices. Therefore, modification of bare electrodes with (hybrid) nanomaterials, especially noble metallic nanoparticles, offers a unique combination of excellent catalytic and sensory properties [65,75,191]. Silver nanoparticles outperform gold and platinum nanomaterials as a low-cost and efficient electrocatalysts for practical sensing applications. Moreover, the synergistic effects of bimetallic [69,86,164], AgNPs/metal oxide composites [109,136] can be also used to promote and/or enhance catalytic sensing.

Due to its diverse structural and morphological forms, carbon is remarkably important and one of the mostly widely used electrode materials [192]. In 54% of the voltammperometric sensors presented in this review, GCE was used as the core material. The main advantages of GCE are its resistance to acidic and alkaline environments and the improved adhesion of the AgNPs to the glassy carbon material [193]. Only a few examples of GCE modified exclusively with AgNPs for the detection of nitrites [65,75,129], nitrobenzene [171,172], and 4-nitrophenol [154] have been found; the vast majority of GCE-based electrochemical sensors reviewed in this manuscript are modified with composite silver (nano)materials. One of the frequently used strategies in the development of novel sensor surfaces is the anchoring of AgNPs on the surface of 2D nanomaterials, mostly graphene-derived, due to their large surface area, high chemical stability, and mechanical strength [9,194]. The GO platform is rich in oxygen-containing functional groups, which greatly improves its hydrophilicity and functionality. Therefore, GO is easily dispersed in both water and organic solvents to obtain a homogeneous dispersion, and can be easily applied to electrode substrates by simple immersion or drop casting technique [130,135]. In addition, GO has an enhanced ability to capture target molecules due to its rich edge oxygen chemistry, a property that has been used to develop a nitrite sensing platform [130]. To increase the sensitivity of the sensor, the strategy of combining AgNPs with a more conductive reduced GO form (restoration of *sp*^2^ hybrid carbon networks) was developed [45,87,155,185]. Thus, (r)GO serves as an outstanding platform for the immobilization of various electroactive species through covalent or noncovalent bonds. In addition to (r)GO sheets, other 2D layered structures with semiconducting properties such as activated C_3_N_4_ [141,170,181], MOFs [46,114], and transition metal dichalogenides [168] have also emerged as prospective candidates for improved catalyst supports for simultaneous voltammetric detection of HMs, selective oxidation of nitrite, amperometric nitrate detection, and selective dopamine quantification, respectively. 3D MWCNTs, especially functionalized by acid treatment, possess excellent thermal and electrical conductivity, which, combined with their large surface area, significantly improves the catalytic properties for the oxidation of ammonium and nitrite species [74,138,144], and the detection of phenolic compounds [80,152] in real water matrices. In only one scientific paper, SWCNTs were used in conjunction with AgNPs as a sensing surface for the quantification of the synthetic estrogen dienestrol in river water [84]. Another approach to increase the sensitivity is to use the CPs in the nanosilver-based sensing hierarchy in the form of an AgNPs@p-1,8-DAN layer [73] or an AgNPs/CB/PEDOT:PSS film [163] built over disc GCE, or ternary hybrid structures of nanosilver on the (r)GO platform with PPy [133] or PANI [134]. Due to their (high) conductivity, redox reversibility, long-term environmental stability, high solution process ability, and simple synthetic procedures with controllable thickness on the sensing electrode, CPs are often employed in voltammperometric sensor design. Synthetic MIPs based on GCE sensing platforms are the best alternative to circumvent the stability and cost issues associated with biological receptors traditionally used for the detection of (bio)molecules [187]. Since the size, shape, and orientation of the recognition sites of the cavities imprinted in MIP directly reflect the properties of the analyte, the molecular imprint technology is more than efficient for the development of highly selective sensors. MIPs in combination with AgNPs and AgNPs/GO composites have proven to be exquisite sensing platforms—picomolar and femtomolar LOD values have been achieved for the detection of the endocrine-disrupting 17-β-estradiol in real water matrices [182,183].

In addition to GCE, another robust carbon-based 3D electrode—CPE—can be quickly fabricated, modified with various (nano)materials, and its surface is easy to clean. Although their sensitivity is (usually) lower than that of GCEs, CPEs decorated with AgNPs have proven to be a reliable tool for amperometric detection of synthetic estrogen (nM LOD) [84], amperometric quantification of nitrites in aquatic solutions (μM LOD) [139], selective voltammetric reduction of *p*-nitroaniline in wastewater (nM LOD) [175], and electrochemical sensing of lead ions in tap water and wastewater specimens (μM LOD) [108]. Another simple and inexpensive form of a carbon-based electrode, with advantages such as high mechanical strength, good quality, stability, and reproducibility, is the PGE [195]. This user-friendly electrode can be used as disposable electrode, eliminating the time-consuming cleaning of solid electrodes. A PGE sensing probe modified with an AgNP-ZnO nanocomposite has been successfully used for the determination of nitrites in lake water and pickled water samples [136], and one decorated with biosynthesized AgNPs was selective towards divalent mercury [97].

Noble metal electrodes, such as gold and platinum material, find application in a variety of electrochemical processes due to their high electrical conductivity, mechanical stability, and chemical resistance. Nevertheless, only a few electrochemical sensors based on nanosilver-decorated noble metal electrodes have been presented in this review. Gold (Au) disc electrodes predominate in the development of voltammetric sensors for the detection of arsenic in river water and lake water [50,119,120], but are also suitable as substrates for the selective and sensitive detection of nitrates in seawater matrices [146,149]. Platinum (Pt) electrodes, decorated with AgNPs biosynthesized utilizing *Agaricus bisporus* mushroom extract [98] and *Mimosa diplotricha* plant leaf extract [99], were found to be an excellent solution for the selective and sensitive detection of toxic Hg(II) in lake water, tap water, and river water specimens, respectively. In addition, a Pt disc electrode decorated with bark green AgNPs (*Moringa oleifera* bark extract) was utilized to detect hazardous Cu(II) from electroplating plant effluents [111], while AgNPs produced with garlic extract were used for the quantification of Cd(II) in lakes [106].

Planar screen-printed (carbon) electrodes are suitable for on-site analyses because they are built on the same substrate with a three-electrode setup (WE, RE, and CE). Planar SPE surfaces can be modified in the same way as disc electrodes by using a: (i) hybrid of AgNPs and Cu(II)–terephthalate MOFs for the quantification of nitrate in drinking water [46], (ii) silver nanoseed/carbon nanofiber modifier for simultaneous detection of Cu(II) and Pb(II) in groundwater matrices [113], (iii) a curcumin-stabilised AgNPs-FeCo_2_O_4_ nanosheets-rGO complex architecture for the detection of 4-NP in industrial wastewater [159], (iv) a ternary Ag/Co_3_O_4_/chitosan composite for the removal of 4-nitrophenol in sewage [160], (v) bimetallic silver–gold–oxide nanoparticles for simultaneous determination of trivalent and hexavalent chromium in tap water [81]. In addition to carbon-based planar surfaces, AgNPs-decorated glass [142], ITO substrates [67,148], and FTO glass substrates [47] were successfully employed for the selective detection of nitrite, nitrate, BPA, and hazardous copper, respectively.

### 4.6. Summary and Future Perspective

In summary, the use of silver nanomaterial-decorated electrodes as voltammperometric sensors offers several advantages based on the electroanalysis output: (i) efficient catalysis; (ii) rapid mass transfer of the target analyte from the bulk solution to the sensing surface and vice versa; (iii) large sensor surface area; and (iv) the ability to precisely tune the electrode microenvironment. Although electrochemical sensing technologies may achieve quantification of analytes even at femtomolar levels, they still lack quality assurance and reliability for long-term measurements in complex water matrices.

Therefore, to ensure an advanced and self-sufficient next generation of nanosilver-based voltammetric/amperometric probes, future research must focus on the: (i) the development of low-cost and submersible sensor platforms for real-time detection of pollutants in water matrices; (ii) fabrication of lighter weight and small-sized devices—miniaturization and planar (all-printed all-solid state) electrode design; (iii) construction of wireless and portable sensors for measurements at remote locations with minor facilities; (iv) sensitivity and selectivity augmentation overcoming interference and fouling challenges; and (v) development of self-sustained devices.

## 5. Conclusions and Outlooks

Electrochemical sensing in combination with nanosilver-based materials and various electroanalytical techniques offers attractive opportunities to meet the requirements for accurate detection of emerging water pollutants. The high conductivity of AgNPs facilitates rapid charge transfer during detection of the target species, while superior catalytic efficiency can be easily tuned by changing the particle size, shape, and temperature of the reaction mixture. Compared to other metal-based nanoparticles, AgNPs have relatively lower cost and toxicity, ensuring their great potential for practical applications. Modification of electrode substrates with silver hybrid architectures by introducing 2D carbon (graphene and its derivatives, carbon black, carbon cloth), 3D carbon nanomaterials (quantum dots, carbon nanotubes), and MOFs or CPs as support materials forms adventurous sensing surfaces for detection of various analytes. Molecular imprinting is by far the most sensitive targeting technology, enabling rapid and ultrasensitive quantification of analytes by fine-tuning the recognition sites in the MIP cavities.

So far, different synthesis strategies have been developed to prepare Ag-based nanocomposite materials decorated WE, from simple immersion and drop casting to in situ reduction method. By combining the advantages of the different synthesis methods and the various modification techniques, high-performance detection is possible for practical in-field applications. Although numerous studies on the synthesis and environmental applications of AgNPs-based electrochemical sensors have been conducted recently, the development of highly selective and sensitive platforms with the possibility of miniaturization and integration into portable devices for real-time measurements and sustainable use of water resources remains a challenge.

## Figures and Tables

**Figure 1 sensors-23-03692-f001:**
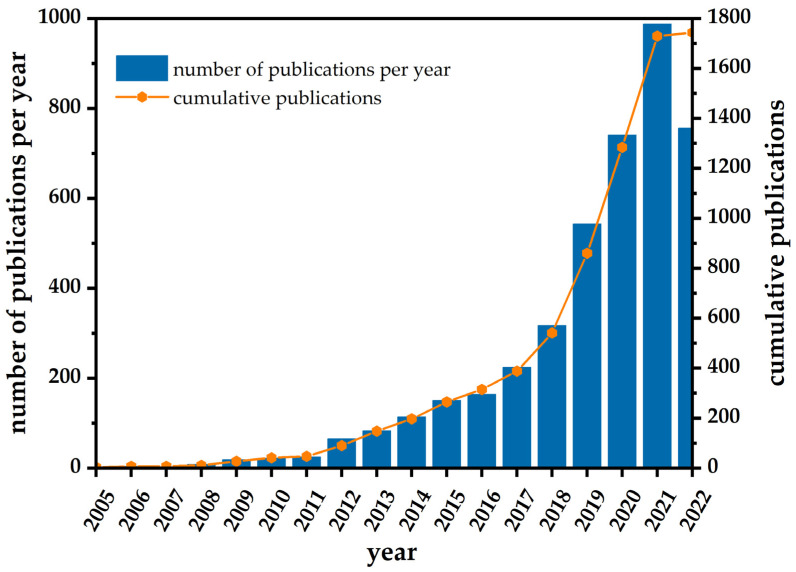
Number of publications per year and cumulative publications regarding nanosilver-based electrochemical sensors for environmental analysis.

**Figure 2 sensors-23-03692-f002:**
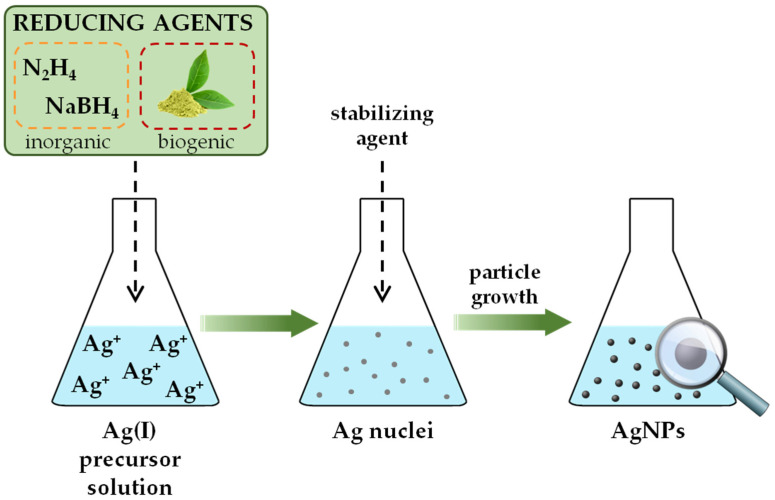
Schematic presentation of AgNP synthesis by reduction in a batch utilizing strong inorganic and mild green reducing agents.

**Figure 3 sensors-23-03692-f003:**
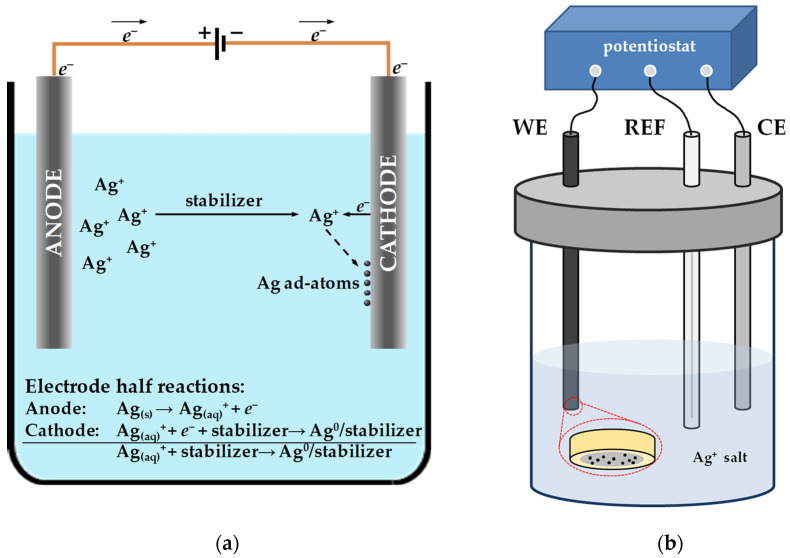
Schematic presentation of an electrochemical synthetic approach in the synthesis of silver nanoparticles using: (**a**) two-electrode electrolytic cell and (**b**) three-electrode electrolytic cell.

**Figure 4 sensors-23-03692-f004:**
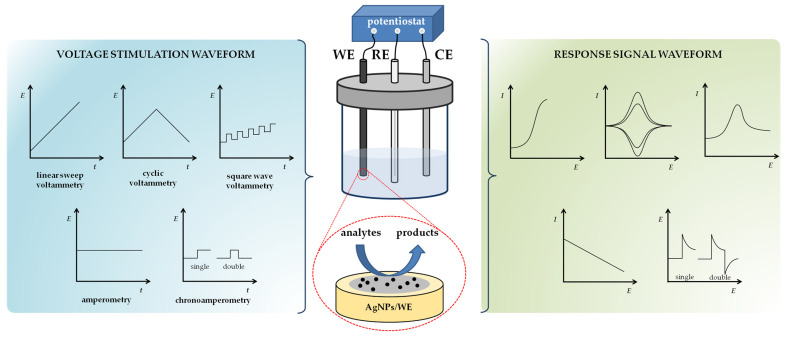
Schematic representation of voltammetric and amperometric techniques used for detection of emerging water pollutants. In the scheme, *E* stands for potential, *I* for current, and *t* for time.

**Figure 7 sensors-23-03692-f007:**
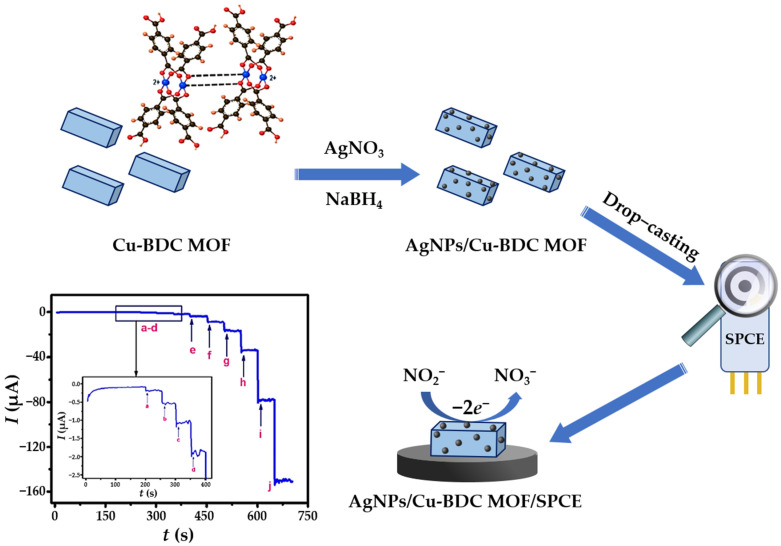
Schematic illustration of silver nanoparticle (AgNPs)-copper(II) terephthalate (Cu-BDC) metal–organic framework (MOF)-modified screen-printed carbon electrode (SPCE) as amperometric sensor for nitrate detection. Adapted with permission from Amali et al. [46]. Copyright 2022 Elsevier.

**Figure 8 sensors-23-03692-f008:**
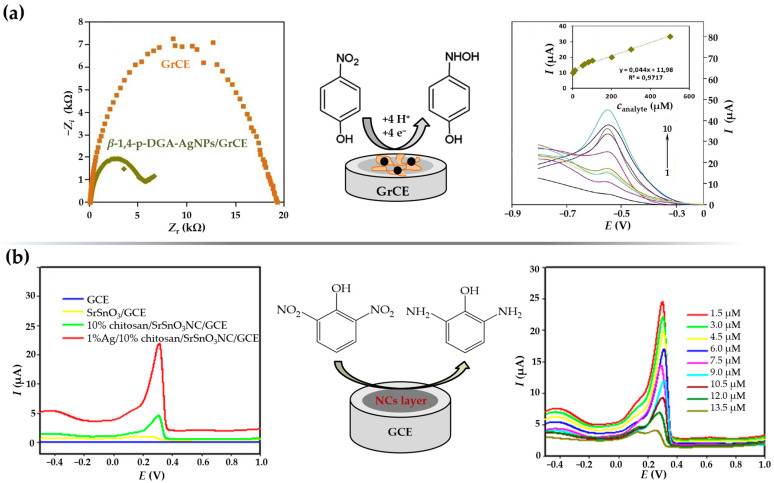
Schematic illustration of the selected voltammetric sensors for the detection of phenolic compounds in water matrices. (**a**) Voltammetric platform based on *β*-1,4-*p*-D-glucosamine-capped AgNPs modified graphite carbon electrode for the detection of 4-NP in wastewater and river water: Nyquist plots for bare and modified GrCE, the mechanism of reduction, and corresponding DPV responses. Reproduced with permission from Ref. [156]; copyright 2023 John Wiley and Sons. (**b**) Ag-chitosan/SrSnO_3_NC/GCE sensor for the voltammetric detection of 2,6-DNP in seawater, underground water, and tap water. Adapted from [161] under the terms of the Creative Commons Attribution (CC BY) licence (https://creativecommons.org/licenses/by/4.0, accessed on 26 February 2023).

**Figure 9 sensors-23-03692-f009:**
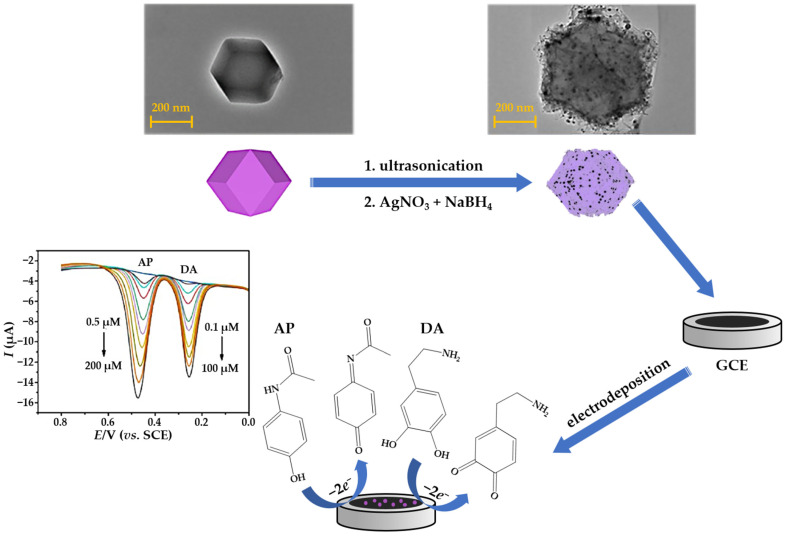
Schematic presentation of the nanosiler-based nanopinnas composite material (Ag-ZIF-67p) DPV sensor for the selective detection of acetaminophen (AP) and dopamine (DA) on the surface of the glassy carbon electrode. Reproduced with permission from ref. [165]. Copyright 2022 Elsevier.

**Figure 10 sensors-23-03692-f010:**
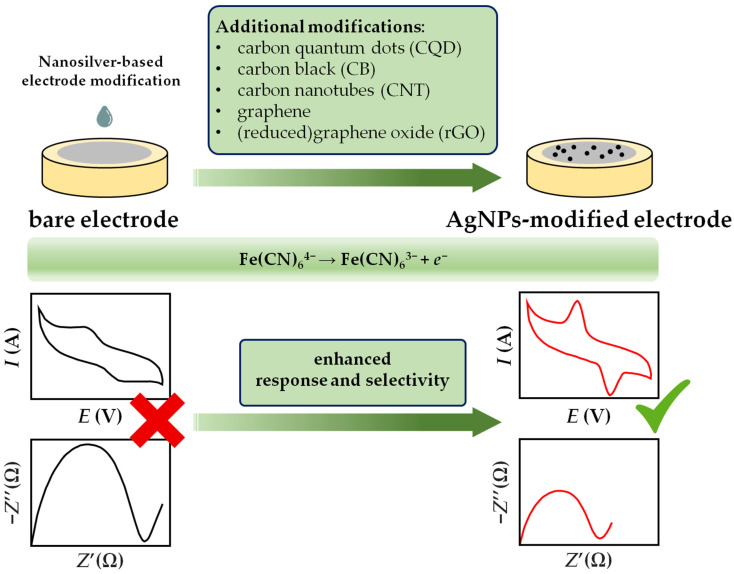
Schematic illustration of AgNPs electrochemical signal boost through the changes in current (CV) and impedance (EIS) responses on bare and nanosilver-modified electrodes.

**Table 4 sensors-23-03692-t004:** Survey of the reviewed silver nanomaterial-based electrochemical sensors for detection of pharmaceuticals in water matrices.

Analyte	Sample	Sensor Design/ Detection Method	Linear Range	AgNP Synthetic Approach	LOD	Ref.
AP	Water, medicine samples	AgNPs-xGnP/GCE/SWV	4.98–33.8 μM	Green synthesis	0.85 nM	[68]
AP LEV	River water	AgNPs-CB-PEDOT:PSS/GCE/ SWV	0.62–7.1 μM 0.67–12 μM	Borohydride reduction	0.012 μM 0.014 μM	[163]
AP	Water samples	CS/Ag-Pd@rGO/GCE/DPV	0.50–300.0 μM	Chemical reduction	0.23 μM	[164]
DA AP	Aqueous pharmaceutical solution	Ag-ZIF-67p/GCE/DVP	0.1–100 μM 0.5–200 μM	Borohydride reduction	0.05 μM 0.2 μM	[165]
AP	Aqueous solution	CNCs@CP5–AgNPs/GCE/DPV	0.5–500 μM	Borohydride reduction	90.0 nM	[166]
DA	Aqueous solution	Ag@MoS_2_/GCE/DPV	1.0–500 μM	Green synthesis	0.2 μM	[168]
DA	Aqueous solution	Pt-Ag/Gr/GCE/DPV	0.1–60 μM	Hydrazine reduction	0.012 μM	[169]

xGnP—exfoliated graphite nanoplatelets; GCE—glassy carbon electrode; SWV—square wave voltammetry; CB—carbon black; PEDOT:PSS—poly(3,4-ethylenedioxytiophene):polystyrene sulfonate; CS/Ag-Pd@rGO—chitosan Ag-Pd bimetallic nanoparticles- reduced graphene oxide; DPV—differential pulse voltammetry; Ag-ZIF-67p—Ag nanoparticles nanopinnas composite material; CNCs©CP5—acid sulfated cellulose nanocrystals cationic pillar[5]arene; Gr—graphene.

**Table 5 sensors-23-03692-t005:** Survey of the reviewed silver nanomaterial-based electrochemical sensors for detection of nitroaromatics in water matrices.

Analyte	Sample	Sensor Design/ Detection Method	Linear Range	AgNP Synthetic Approach	LOD	Ref.
NB	Aqueous solution	EDAS/(g-C_3_N_4_-Ag)/GCE/SWV	5.0–50 μM	Borohydride reduction	2.0 μM	[170]
NB	Aqueous solution	SSG-AuAgNDs/GCE/SWV	1.0–80 μM	Chemical reduction	-	[83]
NB	Tap water, lake water	AgNPs/GCE/DPV	5.0–40 μM	Green synthesis	0.027 μM	[171]
NB	Aqueous solution	AgNPs/GCE/Amp	0.05–21 μM 23–2593 μM	Green synthesis	12.0 nM	[172]
NB	Lake water, tap water, river water, sea water	CC/Ag@GQDs/GCE/DPV	500 nM–1.0 mM	–	30.0 pM	[173]
*p*-NT	Tap water	Ag/AuE/Amp	0.01–0.10 μM	Borohydride reduction	0.092 μM	[174]
*p*-NA	Wastewater	CS-SNPs/CPE/DPV	7.0 nM–1.0 μM	Borohydride reduction	5.0 nM	[175]
NFT 4-NP	Tap water, river water, drinking water	Ag/Se/GCE/DPV	0.1–210 μM 0.1–150 μM	Chemical reduction	23.87 nM 7.82 nM	[176]
AM AT	Tap water, drinking water	COOH-CNTs/Ag/NH_2_-CNT/GCE/SWASV	6 nM–50 pM 9 nM–75 pM	Commercial AgNPs	77.6 fM 83.2 fM	[177]

EDAS—N-[3-(trimethoxysilyl)propyl]ethylenediamine; g-C_3_N_4_—graphitic carbon nitride; GCE—glassy carbon electrode; SWV—square wave voltammetry; SSG—silicate sol–gel matrix; AuAgNDs—bimetallic silver–gold nanodots; DPV—differential pulse voltammetry; Amp—amperometry; CC—carbon cloth; Ag@GQD—silver-graphene quantum dots; AuE—gold electrode; CS-SNPs—chitosan-stabilized silver NPs; CPE—carbon paste electrode; COOH-CNTs—carboxylic acid functionalized carbon nanotubes; NH_2_-CNTs—amino-functionalized carbon nanotubes; SWASV—square wave anodic stripping voltammetry.

## Data Availability

The data presented in this review are available in the manuscript.

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
