# Peer review of "The Role of Silver Nanoparticles in Electrochemical Sensors for Aquatic Environmental Analysis"

_sensors, 2023, doi:10.3390/s23073692_

Round 1
Reviewer 1 Report
The present review article deals with the implementation of silver nanoparticles as electrochemical sensors for environmental purposes. The message is clear, different parts of the review are well organized, the overall work looks well conceived. Pictures are also helping the reader to follow the different examples provided.
Nanoparticles in general represent a hot and very up-to-date topic, in addition environmental analysis is somewhat for which the all community is sensitive for thus the combination of these two topics in one review appears in my opinion very interesting and may trigger new ideas to be developed.
Conclusions seem sound and also references are appropriate.Author Response
The present review article deals with the implementation of silver nanoparticles as electrochemical sensors for environmental purposes. The message is clear, different parts of the review are well organized, the overall work looks well conceived. Pictures are also helping the reader to follow the different examples provided.
Nanoparticles in general represent a hot and very up-to-date topic, in addition environmental analysis is somewhat for which the all community is sensitive for thus the combination of these two topics in one review appears in my opinion very interesting and may trigger new ideas to be developed.
Conclusions seem sound and also references are appropriate.
I would like to thank the reviewer for his/hers constructive and insightful comments on this manuscript. I am very honored to receive such a nice words.
Reviewer 2 Report
The manuscript is an extensive work. By no means can I reject the paper on scientific quality; the author is more versed in this area than myself.
Yet, I would find that nobody would be interested in reading the manuscript. This format is not what I expect from a review article. The manuscript basically summarizes two hundred papers. Namely Section 3. That section is not necessary (supplemental material?). I will probably be the only one ever to read the manuscript from beginning to end.
Well, since there is no political agenda at stake here, and bytes are patient, let it be published.
For the rest I have some minute observations:
The abstract now reads like an introduction. An abstract should summarize the paper and not introduce it.
p.1: When mentioning the hazardous materials, it would be only fair if the author also mentions the hazards of nano-materials (e.g., "understanding the hazards of nanomaterials" by HSE.gov.uk)
No space after a value and %.
"Name, et al." (note the comma).
in references: doi: 10.1060 ... etc. (no need for https:// hyperlink).
p7, l248 "materials"
p7, l269 "three-electrode systems"
p7, eq1 Units are not correct (Paixao also has it like that; remnant from previous times. Here a review paper can help clear it up!)
p8, eq2 idem
p8, Fig4 what is E?
p9 eq4 is redundant (equal to eq3)
p9 eq4 not italics (and both side arrows in chem reactions)
p20 eq5 idem
p9 l363 "coal power plants"
p10, l415 "I-t curve"
p11 l468 "caused"
p11 l469 "containers"
p19 l765 remove "Unfortunately" and "posing a serious problem"
p19 l763 "voltamperometric"
p48 l1867 idem
p49 l1918 idem
p27 l1040 "first"
p29 l1096 "polluting"
p31 l1182 what is Ep?
p39 l1463 "sensor"
p47 l1810 What are MOFs?
p48 l1889 "towards"
Author Response
The manuscript is an extensive work. By no means can I reject the paper on scientific quality; the author is more versed in this area than myself.
Yet, I would find that nobody would be interested in reading the manuscript. This format is not what I expect from a review article. The manuscript basically summarizes two hundred papers. Namely Section 3. That section is not necessary (supplemental material?). I will probably be the only one ever to read the manuscript from beginning to end.
Well, since there is no political agenda at stake here, and bytes are patient, let it be published.
I would like to thank the reviewer for this insightful and constructive critique. I regret that the review of the manuscript was gruelling, I understand that for a reader not versed into the field this manuscript presents a handful, and I am grateful for the time and effort taken. My intention was to provide an in-depth overview of the state-of-the-art in the synthesis and application of AgNPs for environmental purposes (Section 3), as I believe that a thorough and extensive review on the topic lacks in the literature and that it may be of interest to the scientific community. Section 4 discusses the presented scientific papers summarizing them on different basis – from the synthesis of functional material, through development of sensors on different substrates, and mechanisms of quantification of various analytes highlighting the versatile role of AgNPs. Reviewers #1, #3, and #4 have provided positive feedback; therefore, I consider that most interested readers will find the manuscript useful.
For the rest I have some minute observations:
The abstract now reads like an introduction. An abstract should summarize the paper and not introduce it.
The abstract follows a recipe that most abstracts of review papers have, i.e., first to give a small introduction about the importance of the topic, and then a brief dissection of the parts of the manuscript. Since no remark was given on how to rearrange the abstract, and no such comments were received from other reviewers, I have decided to leave the abstract as-is.
p.1: When mentioning the hazardous materials, it would be only fair if the author also mentions the hazards of nano-materials (e.g., "understanding the hazards of nanomaterials" by HSE.gov.uk)
I thank the reviewer for this observation. Since the review is focused on the detection of various groups of organic and inorganic pollutants, their hazard behavior was aligned with the Water Framework Directive (WFD), which prescribes the quality standards for each specimen in environmental and drinking water. I totally agree that silver in its nano-form poses a hazard especially to aquatic organisms. However, since this review highlights the usage of nanosilver as a functional material for the development of voltametric and amperometric sensors, the toxic effects of silver were beyond the scope of this review.
No space after a value and %.
I thank the reviewer for this remark; it has been changed throughout the manuscript.
"Name, et al." (note the comma).
I thank the reviewer for this observation. I have thoroughly revised the manuscript and corrected the missing commas.
in references: doi: 10.1060 ... etc. (no need for https:// hyperlink).
Thank you very much for anticipating this error, it has now been corrected according to your suggestion.
p7, l248 "materials"
Thank you for anticipating this error, it has now been corrected according to your suggestion.
p7, l269 "three-electrode systems"
Thank you very much for this comment, it has now been corrected according to your suggestion.
p7, eq1 Units are not correct (Paixao also has it like that; remnant from previous times. Here a review paper can help clear it up!)
I would like to thank the reviewer for this comment. For clarity, a sentence explaining the units of the constant value has been added in the manuscript (page 7, line 296).
p8, eq2 idem
I would like to thank the reviewer for this comment. Equation 1 (Randles-Ševčik) describes the peak current for the reversible voltametric system (variance of the applied WE potential over a wide potential window), while Equation 2 (Cottrell) describes the change in current with respect to time in a controlled potential experiment (chronoamperometry).
p8, Fig4 what is E?
I would like to thank the reviewer for noticing this ambiguity. For clarity, the description in Fig. 4 has been supplemented with the explanation of the physical quantities.
p9 eq4 is redundant (equal to eq3)
I would like to thank the reviewer for this suggestion. Indeed, at first glance it looks like the same equation is written in reverse, but I would like Eq. 3 and Eq. 4 to remain in this form in order to better emphasize the mechanism of the stripping technique.
p9 eq4 not italics (and both side arrows in chem reactions)
I would like to thank the reviewer again for the correct notation; the italics have been removed, and the both side arrows have been added to both equations.
p20 eq5 idem
Unfortunately, I don't agree that this equation is an idem, as it represents the oxidation mechanism of nitrite species in neutral and slightly acidic media, which corresponds to the pH value of environmental waters, and follows the detection mechanisms of the sensors presented in this review.
p9 l363 "coal power plants"
The word “mining” has now been deleted.
p10, l415 "I-t curve"
I thank the reviewer for noticing this error, it has now been corrected.
p11 l468 "caused"
I thank the reviewer for pointing out this spelling error, which has now been corrected.
p11 l469 "containers"
The word has now been corrected.
p19 l765 remove "Unfortunately" and "posing a serious problem"
The words “unfortunately” and “posing a serious problem” have been removed from the sentence.
p19 l763 "voltamperometric"
The word has been corrected.
p48 l1867 idem
I thank the reviewer for this observation. I realise that at first glance it looks like it is identical, but my intention was to highlight the numerous advantages of CPs in terms of their synthesis, ease to use in the sensor fabrication process, and their impact on improving the analytical features of the sensor.
p49 l1918 idem
I thank the reviewer for this comment, but I must say that I disagree. This section presents the sensor platforms built onto the rigid planar surfaces, introducing firstly screen-printed (carbon) electrodes, and secondly different types of pristine and modified glass supports (ITO and FTO, respectively).
p27 l1040 "first"
The word “firstly” has now been changed to “first”.
p29 l1096 "polluting"
The word “polluted” has now been changed to “polluting”.
p31 l1182 what is Ep?
I would like to thank the reviewer for this excellent observation. In fact, after a thorough revision of the manuscript, I found that this abbreviation had never been explained. Ep stands for the peak potential. Therefore, the sentence has now been rearranged, i.e., the definition of the abbreviation has been inserted.
p39 l1463 "sensor"
I thank you for this comment, the misspelled word is now corrected.
p47 l1810 What are MOFs?
I thank the reviewer for this comment. MOFs are metal-organic frameworks, first introduced within the manuscript on page 14 (line 567). Consequently, only the abbreviation has been used throughout the manuscript thereafter.
p48 l1889 "towards"
The misspelled word has now been changed.
Reviewer 3 Report
This manuscript summarizes the current state of the art and scientific literature on electrochemical sensors based on silver nanomaterials for the detection of different classes of pollutants in aquatic environments, resuming the scientific papers published in the last past 5 years (since 2017).
In the work is reported a consistent introduction about silver nanoparticles. At first the main synthesis techniques for AgNPs are well described dividing them into the two main methodologies: bottom-up and top-down. I found the introduction to be very useful and complete. The use of silver nanoparticles in the voltammetric and amperometric sensors field is then described, demonstrating how their use by electrochemical techniques can be ideal for achieving simple analysis and high sensitivity. In the various sections, the advantages of their use are well underlined: fast responses, low detection limits, ease of use and possibility of analysis in situ. Uses for monitoring different metals in water are reported, with particular attention on heavy metals.
The sections are very well organized ensuring a linear drafting of the work, allowing the reader to easily understand the research and the state of art about AgNPs. In addition, the manuscript has been written in a concise manner.
Author Response
This manuscript summarizes the current state of the art and scientific literature on electrochemical sensors based on silver nanomaterials for the detection of different classes of pollutants in aquatic environments, resuming the scientific papers published in the last past 5 years (since 2017).
In the work is reported a consistent introduction about silver nanoparticles. At first the main synthesis techniques for AgNPs are well described dividing them into the two main methodologies: bottom-up and top-down. I found the introduction to be very useful and complete. The use of silver nanoparticles in the voltammetric and amperometric sensors field is then described, demonstrating how their use by electrochemical techniques can be ideal for achieving simple analysis and high sensitivity. In the various sections, the advantages of their use are well underlined: fast responses, low detection limits, ease of use and possibility of analysis in situ. Uses for monitoring different metals in water are reported, with particular attention on heavy metals.
The sections are very well organized ensuring a linear drafting of the work, allowing the reader to easily understand the research and the state of art about AgNPs. In addition, the manuscript has been written in a concise manner.
I would like to thank the esteemed reviewer for the detailed comments on this manuscript, and for acknowledging the effort that was required to present this scientifically sound topic as comprehensively as possible. I am very honored to receive such kind words.
Reviewer 4 Report
1. In the tables take care of the same number of decimal places.
Good review!
Author Response
- In the tables take care of the same number of decimal places.
Good review!
First of all, I would like to thank the esteemed reviewer for this short but very meaningful comment, that said everything in a simple way. It is an honor to receive such wonderful words.
I would also like to thank you for pointing out the problem of decimal places. Since these data were taken from the original scientific papers, those that do not change the context were modified, but those that unfortunately could not be written differently were left in their original form.